# Decentralized SGD with Controlled Disagreement Finds Flatter Minima

## Abstract

Decentralized training is often regarded as inferior to centralized training because the consensus errors between workers are thought to undermine convergence and generalization. This work challenges this view by introducing decentralized SGD with Adaptive Consensus (DSGD-AC), which uses a learning-rate-synchronized scaling mechanism to maintain consensus errors during the learning-rate decay. Under a frozen local window linearization, we show that adaptive consensus changes the steady-state variance of disagreement modes by balancing two effects: it preserves consensus-error magnitude through weaker graph damping while still allowing curvature-dependent damping to shape the disagreement directions. This balance can produce a stronger Hessian-weighted loss-envelope penalty around the deployed model than standard DSGD, which is associated with flatter solutions in our experiments. Empirical results on image classification show that DSGD-AC reaches flatter solutions and higher test accuracy than standard DSGD and even centralized SGD. Together, these results support consensus errors as a useful implicit regularizer and open a new perspective on the design of decentralized learning algorithms.

## 1 Introduction

In large-scale deep learning, decentralized training lets workers exchange model parameters only with neighbors, avoiding the cost of global synchronization and global all-reduce communication (Abadi et al., 2016; Li et al., 2020). This reduces latency and eliminates single points of failure, making decentralized approaches attractive for geographically distributed systems (Dhasade et al., 2023; Gholami & Seferoglu, 2024) and GPU clusters (Lian et al., 2017; Assran et al., 2019; Wang et al., 2025).

Despite its practical appeal, decentralized training methods such as the decentralized stochastic gradient descent (DSGD) trail centralized training in terms of convergence and generalization, even when data distributions of workers are i.i.d. (Koloskova et al., 2020; Zhu et al., 2022; Yuan et al., 2023). This gap was largely attributed to consensus errors, the persistent discrepancies in the model parameters between different workers (Alghunaim & Yuan, 2022; Zhu et al., 2022). Prior work has therefore focused on reducing these errors through improved communication topologies (Ying et al., 2021; Takezawa et al., 2023) and algorithm designs (Pu & Nedić, 2021; Wang et al., 2019; Lin et al., 2021) to make decentralized training closer to its centralized counterpart.

However, this perspective neglects the potential constructive role of consensus errors. Rather than acting as detrimental noise, these discrepancies may serve as structured perturbations that encourage preference for flatter minima in the loss landscape, solutions that strongly correlate with better generalization (Jiang et al., 2019). This view draws inspiration from sharpness-aware minimization strategies (Foret et al., 2020; Bisla et al., 2022; Li et al., 2024b), which improve model robustness and generalization by explicitly introducing curvature-aware perturbations.

In this study, we challenge the conventional view by introducing Decentralized SGD with Adaptive Consensus (DSGD-AC), an algorithm that strategically maintains consensus errors through a time-varying and learning-rate-synchronized scaling mechanism. Based on the local linearized model, our analysis characterizes the steady-state variance of disagreement modes and shows how the adaptive consensus factor balances the

magnitude and curvature profile of consensus errors, inducing a stronger Hessian-weighted penalty around the deployed global average. A radius-matched isotropic perturbation control separates these effects: matching the perturbation magnitude recovers part of the generalization gain, while the flatter DSGD-AC solution indicates that the structured perturbation directions carry an effect of their own. We empirically verify the theory with comprehensive numerical results and diagnostics on Hessian information. DSGD-AC incurs negligible additional computational cost relative to standard SGD or DSGD, while retaining the communication efficiency and runtime advantages of decentralized training.

The main contributions of this work are threefold. First, we propose DSGD-AC, an adaptive consensus algorithm that maintains theoretically motivated consensus errors, empirically finds flatter minima, and improves generalization on deep learning tasks at minimal computational cost. Second, we provide theoretical analysis and empirical diagnostics of the modal variance and consensus error radius based on a local frozen linearization, showing that the controlled errors induce a stronger Hessian-weighted penalty and are associated with flatter solutions. Third, we evaluate DSGD-AC across models, worker counts, and communication topologies, including comparisons with the consensus-promoting algorithm and explicit sharpness-aware training, and characterize its performance–cost trade-off.

### 1.1 Related work

**Canonical view of consensus errors** The prevailing view in decentralized training is that models should minimize the consensus errors and approximate centralized training as closely as possible. To mitigate discrepancies among local models caused by weakly connected networks, prior work has focused on tracking global information (Wang et al., 2019; Pu & Nedić, 2021; Yuan et al., 2021; Takezawa et al., 2022), enhancing communication topologies to improve convergence rates (Ying et al., 2021; Zhu et al., 2022; Takezawa et al., 2023), and more. In addition, several theoretical studies (Zhu et al., 2022; Alghunaim & Yuan, 2022) establish a theoretical connection between the connectivity of decentralized communication topologies and both convergence and generalization, demonstrating that weaker connectivity results in poorer outcomes on both fronts. In contrast, we demonstrate the constructive role of consensus errors by showing that controlled disagreement can induce a meaningful Hessian-weighted loss-envelope penalty, and we propose DSGD-AC as a practical way to exploit this effect in deep learning tasks.

**Explorations beyond the canonical view** Since the canonical perspective dominates, research into the potential benefits of consensus errors remains sparse. Kong et al. (2021) empirically identified error thresholds and noted advantages in specific training phases; however, they did not explore regimes where errors exceed those of ring-topology DSGD, and their control scheme showed limited gains. Zhu et al. (2023) offer a novel interpretation, framing consensus errors in DSGD as random perturbations around the global average, which are asymptotically equivalent to average-direction sharpness-aware minimization (SAM) (Bisla et al., 2022). Our work further identifies how the magnitude and Hessian-weighted effect of consensus errors are shaped by gradient noise and curvature-dependent damping, and, by proposing DSGD-AC, shows how decentralized training can use disagreement constructively beyond the large-batch setting.

**Explicit curvature-related perturbations** If the global average is taken as the deployed model (Zhu et al., 2023), decentralized learning can be interpreted as training the average model, and worker deviations act as perturbations around it. Sharpness-aware minimization (SAM) was first proposed by Foret et al. (2020) to improve the generalization of deep neural networks, and many variants (Kwon et al., 2021; Bisla et al., 2022; Liu et al., 2022; Li et al., 2024a; Luo et al., 2024) were developed to improve it further. However, to achieve the best performance, these algorithms typically require additional gradient evaluations and increase the computational cost significantly. Our work uses consensus errors as free perturbations that can enhance generalization without introducing extra computation or memory beyond DSGD.

## 2 Problem setting and notation

We consider decentralized training in a data center where the full dataset is accessible to all workers. The training uses a standard distributed data sampler: at the beginning of each epoch, the dataset is reshuffled

and evenly partitioned across workers, yielding i.i.d. data distributions in expectation. This setting corresponds to the standard decentralized optimization regime studied by Assran et al. (2019); Ying et al. (2021); Kong et al. (2021); Zhu et al. (2023); Wang et al. (2025).

**Decentralized optimization** We consider $n$ workers collaborating to train a $d$-dimensional model. Let $[n]$ denote the set of integers $\{1, 2, \ldots, n\}$. Each worker $i \in [n]$ holds a local objective determined by its local dataset $\mathcal{D}_i$:

$$f_i(x) = \mathbb{E}_{s \sim \mathcal{D}_i}[f_i(x; s)]. \tag{1}$$

Under the i.i.d. setting considered here, all workers share the same underlying objective but operate on different mini-batches drawn from the shared dataset. Each worker maintains its own local model $x_i$. The workers collaboratively solve

$$\begin{aligned}
\underset{\{x_1, x_2, \cdots, x_n\}}{\text{minimize}} \quad & F(x_1, \cdots, x_n) = \tfrac{1}{n} \sum_{i=1}^{n} f_i(x_i), \\
\text{subject to} \quad & x_i = x_j, \qquad \forall i, j \in [n],
\end{aligned} \tag{2}$$

where the constraint enforces consensus across workers.

**Decentralized SGD (DSGD)** The update of DSGD (Lian et al., 2017) on worker $i$ is:

$$x_i^{(t)} = x_i^{(t-1)} - \alpha^{(t)} \nabla f(x_i^{(t-1)}; s_i^{(t)}) + \sum_{j \in \mathcal{N}(i)} W_{ij}(x_j^{(t-1)} - x_i^{(t-1)}) \tag{3}$$

where $\mathcal{N}(i)$ is the neighbor set of worker $i$ (including itself), $W$ is a symmetric, non-negative, doubly stochastic matrix defining the weights of the edges with $W_{ij} = 0$ if $j \notin \mathcal{N}(i)$), and $s_i^{(t)}$ is the mini-batch sampled by worker $i$ at iteration $t$.

Following standard notation, we denote the global average by $\bar{x}^{(t)} := \tfrac{1}{n} \sum_{i=1}^{n} x_i^{(t)}$, the consensus errors $\delta_i^{(t)} := x_i^{(t)} - \bar{x}^{(t)}$ and the matrix forms $X^{(t)} := [x_1^{(t)}, \cdots, x_n^{(t)}]^\top$, $G^{(t)} := [g_1^{(t)}, g_2^{(t)}, \cdots, g_n^{(t)}]^\top$, $\Delta^{(t)} = [\delta_1^{(t)}, \cdots, \delta_n^{(t)}]^\top$, and $\bar{X}^{(t)} = [\bar{x}^{(t)}, \cdots, \bar{x}^{(t)}]^\top$. We also denote the center projector $P = I - \tfrac{1}{n} \mathbf{1}\mathbf{1}^\top$ and the graph Laplacian $L = I - W$. Then, the DSGD update in Eq. (3) can be written in matrix form

$$X^{(t)} = X^{(t-1)} - \alpha^{(t)} G^{(t)} + (W - I) X^{(t-1)} \tag{4}$$

We focus on the adapt-while-communication variant, which has been shown to be more runtime efficient (Lian et al., 2017; Assran et al., 2019; Wang et al., 2025) and to share the same generalization bound as the adapt-then-communication variant (Bellet et al., 2023).

## 2.1 Assumptions

Our theoretical analysis relies on the following assumptions.

**Assumption 1** (Twice-differentiable objectives)**.** *The objective functions $f_i$ are at least twice differentiable.*

**Assumption 2** (Decentralized communication topology)**.** *The graph is connected and undirected. $W$ is non-negative, symmetric, and doubly stochastic.*

**Assumption 3** (I.i.d. data distributions)**.** *All workers have access to the full dataset, so that*

$$f_1 = f_2 = \cdots = f_n = f$$

**Assumption 4** (Martingale-difference gradient noise)**.** *Let $g_i^{(t)}$ be the mini-batch gradient on worker $i$. Under $f_i = f$, its first-order expansion around $\bar{x}^{(t-1)}$ is*

$$g_i^{(t)} = \nabla f(\bar{x}^{(t-1)}) + H^{(t)} \delta_i^{(t-1)} + r_i^{(t)} + \xi_i^{(t)},$$

*where $H^{(t)} = \nabla^2 f(\bar{x}^{(t-1)})$ is the Hessian, $r_i^{(t)}$ the Taylor residual, and $\xi_i^{(t)}$ mini-batch noise. In matrix form,*

$$G^{(t)} = \mathbf{1} \nabla f(\bar{x}^{(t-1)})^\top + \Delta^{(t-1)} H^{(t)} + R^{(t)} + \Xi^{(t)}.$$

Let $\mathcal{F}_{t-1}$ be the information available before sampling the mini-batches at iteration $t$. The noise is conditionally mean zero:

$$\mathbb{E}\left[\Xi^{(t)} \mid \mathcal{F}_{t-1}\right] = 0$$

**Assumption 5** (Frozen local stochastic window)**.** *For the modal analysis, we consider a frozen approximation of a short late-training window in which $H^{(t)} = H_f$, $R^{(t)} = 0$, $\alpha^{(t)} = \alpha$, and $\gamma^{(t)} = \gamma$. The conditional second moment of the projected gradient noise is finite and constant in this frozen model:*

$$\mathbb{E}\left[\mathrm{vec}(P\Xi^{(t)})\,\mathrm{vec}(P\Xi^{(t)})^\top \mid \mathcal{F}_{t-1}\right] = \Sigma_f$$

*All modal variance and radius statements below refer to steady-state second moments of the frozen local model.*

## 3 DSGD-AC: preserving disagreement under learning-rate decay

Decentralized SGD couples two opposing forces: local updates create disagreement among workers, whereas parameter mixing removes it. As the learning rate decays, the local updates weaken while the mixing operator continues to contract disagreement, progressively drawing the worker models toward their average. Since the worker deviations also act as perturbations around that average, their late-training collapse may remove an important regularization effect of decentralized optimization.

This section characterizes the disagreement collapse in DSGD, introduces a learning-rate-synchronized consensus rule that prevents it, and explains through a local modal analysis how graph mixing and curvature shape the retained deviations. Throughout the section, our running example trains Wide ResNet (WRN)-16-8 (Zagoruyko & Komodakis, 2016) on CIFAR-100 (Krizhevsky et al., 2009) with 8 workers and a one-peer ring topology under cosine annealing with linear warm-up. Detailed settings are given in Appendix A.3.

### 3.1 Learning-rate decay collapses disagreement in DSGD

Let $\delta_i^{(t)} = x_i^{(t)} - \bar{x}^{(t)}$ denote worker $i$'s deviation from the deployed average model. Prior work interprets these deviations as perturbations around $\bar{x}^{(t)}$ and relates DSGD asymptotically to average-direction SAM (Zhu et al., 2023; Bisla et al., 2022). Unlike SAM, which explicitly fixes a perturbation radius, DSGD generates its perturbations through stochastic local updates. Their scale therefore contracts with $\alpha^{(t)}$, and standard convergence analyses accordingly predict vanishing consensus error under decaying step sizes (Lian et al., 2017; Zhu et al., 2022). Figure 1b shows this collapse in the running example.

The per-step surrogate associated with the DSGD update makes this interaction transparent:

$$J^{(t)}(x_1, \ldots, x_n) = \underbrace{\sum_{i=1}^n f_i(\bar{x}^{(t)})}_{\text{objective at deployed model}} + \underbrace{\sum_{i=1}^n \left[f_i(x_i^{(t)}) - f_i(\bar{x}^{(t)})\right]}_{\text{sharpness}} + \underbrace{\frac{1}{4\alpha^{(t)}} \sum_{i,j \in [n]} W_{ij} \|x_i^{(t)} - x_j^{(t)}\|^2}_{\text{consensus regularizer}}. \tag{5}$$

Evaluating the gradient of the corresponding per-step expression at the previous iterate recovers the DSGD update in Eq. (3) (derivation in Appendix A.1.1). As $\alpha^{(t)}$ decreases in standard DSGD, the coefficient $1/(4\alpha^{(t)})$ in Eq. (5) grows, while the stochastic updates that sustain disagreement become smaller. The resulting contraction drives $\delta_i^{(t)}$ toward 0 and voids the potential sharpness regularization. Maintaining a non-vanishing consensus-error radius throughout training is therefore necessary to preserve the potential regularization effect.

A constant-step control confirms that learning-rate decay drives this collapse: standard DSGD retains substantial disagreement without decay but performs markedly worse on the test set, making this a diagnostic rather than a competitive schedule in this setup (Appendix A.2.7).

### 3.2 DSGD-AC: learning-rate-synchronized consensus scaling

DSGD-AC addresses this collapse by scaling the mixing step with a time-dependent factor $\gamma^{(t)}$. The algorithm is detailed in Algorithm 1. In terms of the surrogate $J^{(t)}$ in Eq. (5), this changes the consensus-regularizer

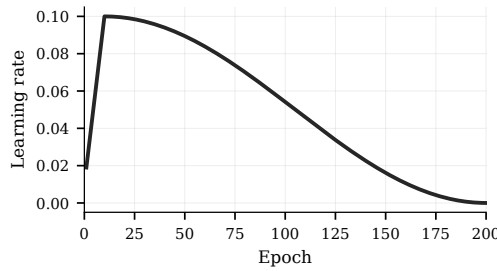

(a) Cosine annealing learning rate schedule

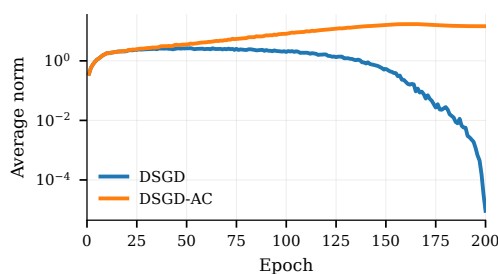

(b) Norm of consensus errors evaluated across epochs

Figure 1: Decentralized training of WRN16-8 on CIFAR-100 with 8 workers and the one-peer ring topology.

weight to $\gamma^{(t)}/(4\alpha^{(t)})$, preventing it from dominating as $\alpha^{(t)} \to 0$. The modified lines are highlighted in Algorithm 1, and the algorithm takes the global average $\bar{x}^{(t)}$ as the deployed model.

---

**Algorithm 1:** Decentralized SGD with adaptive consensus (DSGD-AC) on worker $i$

---

**Data:** Dataset $(D)$, the number of workers $(n)$, the number of epochs $(E)$, the number of batches per epoch $(T)$, initialization $(x^{(0)})$, and hyperparameters $(p \in \mathbb{R}^+$ and $E_{\text{start}} \in [E])$ .

**Result:** Deployed model $\bar{x} = \frac{1}{n}\sum_{j=1}^{n} x_j^{(TE)}$

$x_1^{(0)} = x_2^{(0)} = \cdots = x_n^{(0)} = x^{(0)}$

**for** $t = 1$ **to** $TE$ **do**

 $g_i^{(t)} = \nabla f(x_i^{(t-1)}; s_i^{(t)})$

 $\gamma^{(t)} = \begin{cases} 1, & \lceil t/T \rceil < E_{\text{start}}, \\ \left[\alpha^{(t)}/\alpha_{\max}\right]^p, & \lceil t/T \rceil \geq E_{\text{start}}. \end{cases}$

 $x_i^{(t)} = x_i^{(t-1)} - \alpha^{(t)}g_i^{(t)} + \gamma^{(t)} \sum_{j \in \mathcal{N}(i)} W_{ij}(x_j^{(t-1)} - x_i^{(t-1)})$

**end**

---

Note that $\alpha^{(t)}$ is determined by a learning rate scheduler such as one in Figure 1a, and $\alpha_{\max}$ is the maximum learning rate over the iterations in which adaptive consensus is active. The start epoch $E_{\text{start}}$ delays activation: DSGD-AC uses $\gamma^{(t)} = 1$ and therefore reduces to standard DSGD before epoch $E_{\text{start}}$, then applies the learning-rate-synchronized scaling rule from that epoch onward. This normalization ensures $\gamma^{(t)} \in [0, 1]$ and prevents a consensus step stronger than the base DSGD mixing step. The exponent $p$ controls how aggressively mixing is relaxed during learning-rate decay. We use $p = 3$ and $E_{\text{start}} = 10$ (after the learning rate warmup phase) in the showcase experiments, which are chosen based on hyperparameter tuning.

The term adaptive refers specifically to the adaptation to the current value of a given learning-rate schedule, rather than feedback based on an observed training statistic. DSGD-AC is therefore designed for the decaying-learning-rate regime, where standard DSGD loses its late-training disagreement.

Figure 1b confirms that DSGD-AC prevents the late-training collapse of disagreement. The experiments in Section 4 show that DSGD-AC consistently improves test accuracy and test loss and reaches solutions with smaller largest Hessian eigenvalues than standard DSGD across various models and distributed setups.

### 3.3 Radius, direction, and curvature exposure

The effect of disagreement depends jointly on its magnitude and direction. The loss envelope around the deployed average model $\bar{x}^{(t)}$ makes this interaction explicit by separating the effect of the worker deviations

$\delta_i^{(t)}$ from the objective value at the average. A second-order expansion gives

$$\sum_{i=1}^{n} f(\bar{x}^{(t)} + \delta_i^{(t)}) \approx n f(\bar{x}^{(t)}) + \frac{1}{2} \sum_{i=1}^{n} (\delta_i^{(t)})^{\top} H^{(t)} \delta_i^{(t)}, \tag{6}$$

where the first-order terms cancel because $\sum_i \delta_i^{(t)} = 0$, and the quadratic term is a Hessian-weighted penalty on the consensus errors, concentrated on positive-curvature directions. Although the remainder is not necessarily negligible when the consensus-error radius is not close to zero, the approximation captures the dominant components, and we evaluate the scope of this approximation empirically in Appendix A.2.1.

To empirically assess the interaction between the consensus errors and curvature, we track four quantities defined in terms of the Hessian $H^{(t)} := \nabla^2 f(\bar{x}^{(t)})$, its largest eigenvalue $\lambda_1(H^{(t)})$, and the consensus errors $\Delta^{(t)}$. The first diagnostic is the curvature exposure $Q_t$, the Hessian-weighted consensus-error penalty from the loss envelope in Eq. (6):

$$Q_t := \sum_{i=1}^{n} \frac{(\delta_i^{(t)})^{\top} H^{(t)} \delta_i^{(t)}}{\lambda_1(H^{(t)})}, \quad \mathcal{E}_t := Q_t \lambda_1(H^{(t)}), \tag{7}$$

where $\mathcal{E}_t$ is an auxiliary variable for the unnormalized curvature exposure.

The second is the normalized alignment score, separating the directional component of $Q_t$ from its radius:

$$A_t := \frac{\operatorname{tr}\left(\widetilde{\Delta}^{(t)} H^{(t)} (\widetilde{\Delta}^{(t)})^{\top}\right)}{\lambda_1(H^{(t)})}, \tag{8}$$

Here, $\widetilde{\Delta}^{(t)} := \Delta^{(t)} / \|\Delta^{(t)}\|_F$ so $Q_t = \|\Delta^{(t)}\|_F^2 \cdot A_t$, decomposing the penalty into radius and alignment. The third is the random-direction baseline, the expected normalized alignment of Rademacher directions,

$$B_t := \frac{\mathbb{E}_u\left[u^{\top} H^{(t)} u / \|u\|^2\right]}{\lambda_1(H^{(t)})} = \frac{\operatorname{tr}(H^{(t)})}{d \cdot \lambda_1(H^{(t)})}, \tag{9}$$

estimated using 50 Hutchinson trace probes. Together, $Q_t$, $A_t$, and $B_t$ diagnose how the consensus errors interact with the local curvature at the deployed model; all three are normalized by $\lambda_1(H^{(t)})$ to remove loss-scaling bias and facilitate comparison across runs. Further evaluation details are in Appendix A.3.3.

Figure 2 separates the magnitude and direction components of the DSGD – DSGD-AC comparison. Standard DSGD exhibits substantial disagreement early in training and often stronger normalized alignment $A_t$ than DSGD-AC. At a matched radius, this higher $A_t$ would make DSGD's deviations more curvature-efficient. Its actual radius, however, collapses as the learning rate decays, forcing $Q_t$ to collapse through $Q_t = \|\Delta^{(t)}\|_F^2 \cdot A_t$. DSGD-AC has somewhat weaker normalized alignment, but its substantially larger radius more than compensates, producing greater total curvature exposure through the middle and late stages. For both methods, $A_t$ remains above the corresponding isotropic reference $B_t$ during late training, so the surviving disagreement is directionally structured rather than arbitrary noise. DSGD-AC therefore does not improve by maximizing normalized alignment; it preserves a sufficiently large radius while retaining meaningful directional structure.

These diagnostics explain why DSGD-AC maintains greater curvature exposure than standard DSGD, but they do not determine whether preserving the larger radius alone would produce the same optimization effect. To isolate the contribution of perturbation magnitude from that of direction, we introduce ISO-SGD as a diagnostic, radius-matched isotropic control. ISO-SGD follows the synchronous-SGD setup and uses independent isotropic Gaussian perturbations for each worker, and the norm at each epoch is set to the average worker consensus-error norm measured from the DSGD-AC reference run at that epoch. It therefore reproduces the practical perturbation-radius schedule while removing the directional structure generated by decentralized training. The detailed construction is deferred to Appendix A.3.3.

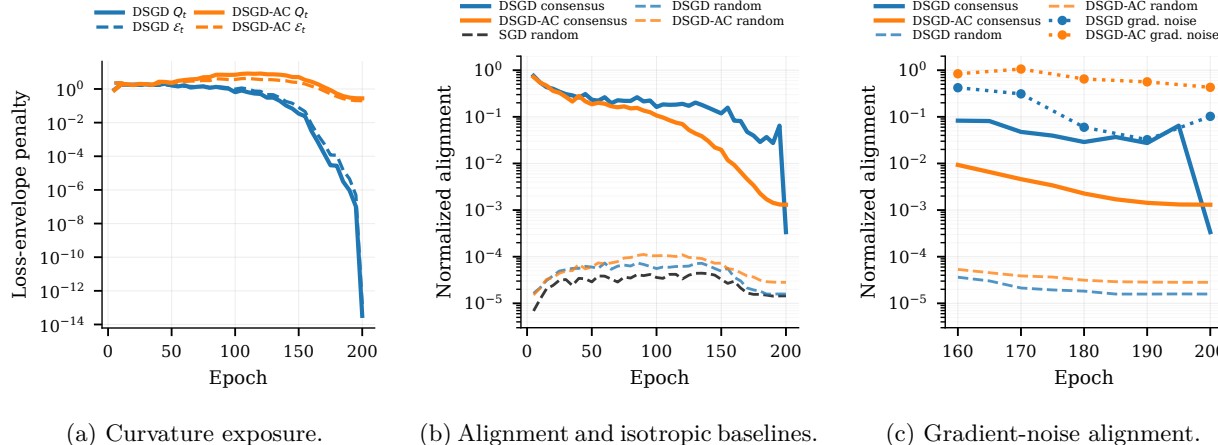

(a) Curvature exposure.  (b) Alignment and isotropic baselines.  (c) Gradient-noise alignment.

Figure 2: Hessian diagnostics for WRN16-8 on CIFAR-100: (a) normalized and unnormalized curvature exposure for DSGD and DSGD-AC; (b) consensus-error alignment for DSGD and DSGD-AC together with isotropic baselines evaluated at the SGD, DSGD, and DSGD-AC solutions; and (c) late-training gradient-noise alignment. Except for $\mathcal{E}_t$, all quantities are normalized by the largest Hessian eigenvalue.

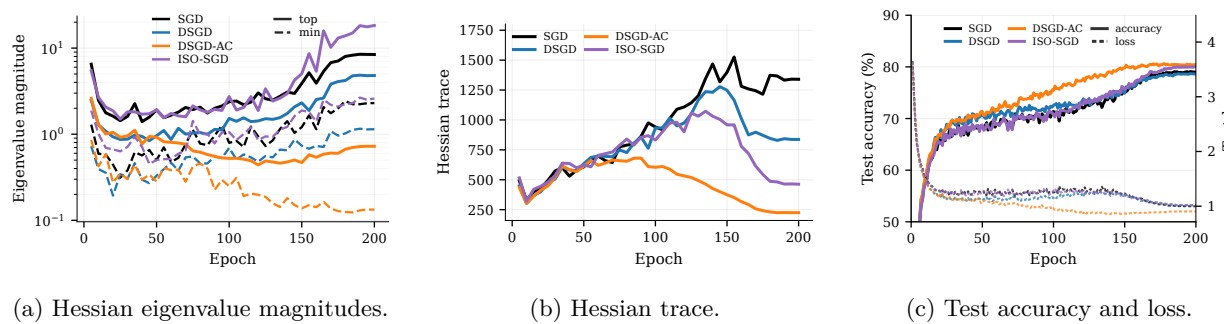

(a) Hessian eigenvalue magnitudes.  (b) Hessian trace.  (c) Test accuracy and loss.

Figure 3: Training WRN16-8 on CIFAR-100 with 8 workers and the one-peer ring topology with synchronous SGD, DSGD, DSGD-AC, and ISO-SGD. ISO-SGD improves over SGD and DSGD in final accuracy, but DSGD-AC reaches the highest accuracy and the flattest local region according to both the extreme eigenvalues and the Hessian trace. Panel (c) shows test accuracy on the left axis and test loss on the right axis. The extreme eigenvalues are estimated by 30 Lanczos iterations.

ISO-SGD improves final accuracy over SGD and DSGD, recovering a substantial part of the gain achieved by DSGD-AC. However, it converges to a substantially sharper solution than DSGD-AC according to both the largest Hessian eigenvalue and the Hessian trace, and it remains slightly less accurate on the test set. The radius-matched control therefore shows that perturbation magnitude is important but does not fully explain the DSGD-AC result: the structured directions provide an additional benefit beyond radius alone. Figures 2 and 3 assign distinct roles to magnitude and direction: the radius preservation explains why DSGD-AC maintains curvature exposure after standard DSGD collapses, while the ISO-SGD comparison establishes that radius alone is not the complete explanation.

## 3.4 Local modal explanation of controlled disagreement

The preceding diagnostics establish that DSGD-AC preserves curvature exposure through a larger, directionally structured disagreement radius, but they do not explain how weakening consensus produces this behavior. We study that mechanism using a frozen late-training approximation in which graph mixing and Hessian curvature provide distinct sources of modal damping.

We start from the matrix form of DSGD-AC, analogous to Eq. (4):

$$
\begin{aligned}
X^{(t)} &= X^{(t-1)} - \alpha^{(t)} G^{(t)} + \gamma^{(t)} (W - I) X^{(t-1)} \\
&= (I - \gamma^{(t)} L) X^{(t-1)} - \alpha^{(t)} G^{(t)}
\end{aligned}
\tag{10}
$$

Multiplying by the center projector $P$ removes the average model and isolates the disagreement $\Delta^{(t)} = P X^{(t)}$. Since $W$ is doubly stochastic under Assumption 2, $PL = LP$ and we find the disagreement recursion

$$
\Delta^{(t)} = P(I - \gamma^{(t)} L) X^{(t-1)} - \alpha^{(t)} P G^{(t)} = (I - \gamma^{(t)} L) \Delta^{(t-1)} - \alpha^{(t)} P G^{(t)}
\tag{11}
$$

Equation (11) already shows the two competing forces: mixing damps disagreement through $\gamma^{(t)} L$, while the worker-wise stochastic gradients continually inject disagreement through $P G^{(t)}$. To see which Hessian directions receive this injected energy, we linearize the gradients around the deployed average model and diagonalize both the graph Laplacian and the Hessian.

**Lemma 1** (Linearized modal dynamics). *Under Assumptions 1–5, the local linearized dynamics in a late-training window decouple exactly into scalar recursions. Let $L = V \operatorname{diag}(\mu_1, \ldots, \mu_n) V^\top$ with $2 \geq \mu_1 \geq \cdots \geq \mu_{n-1} > \mu_n = 0$, and let $H_f = U \operatorname{diag}(\lambda_1, \ldots, \lambda_d) U^\top$ with $\lambda_1 \geq \cdots \geq \lambda_d$. Define $z_{j,k}^{(t)} = v_j^\top \Delta^{(t)} u_k$ and $\xi_{j,k}^{(t)} = v_j^\top \Xi^{(t)} u_k$. Then each graph-Hessian mode satisfies*

$$
z_{j,k}^{(t)} = (1 - \gamma \mu_j - \alpha \lambda_k) z_{j,k}^{(t-1)} - \alpha \xi_{j,k}^{(t)}.
\tag{12}
$$

The scalar recursion in Lemma 1 offers a local explanation for DSGD-AC and explains many of the empirical observations in Section 3.3. The effective damping of mode $(j, k)$ is $d_{j,k} := \gamma \mu_j + \alpha \lambda_k$, the sum of a graph term and a curvature term. Decreasing $\gamma$ weakens only the graph damping, while the positive Hessian curvature continues to suppress sharp directions independently. Theorem 1 makes this precise by computing the steady-state variance of each mode as a function of $d_{j,k}$, the noise variance $\sigma_{j,k}^2$, and the learning rate $\alpha$.

**Theorem 1** (Steady-state modal variance). *Under Assumptions 1–5, define*

$$
\sigma_{j,k}^2 := \mathbb{E}\left[ \left( \xi_{j,k}^{(t)} \right)^2 \mid \mathcal{F}_{t-1} \right].
$$

*For every disagreement mode $j \in \{1, \ldots, n-1\}$ satisfying the stability condition $0 < d_{j,k} < 2$, the second moment converges from any finite initialization to the unique steady-state value*

$$
\mathbb{E}[z_{j,k}^2] = \frac{\alpha^2 \sigma_{j,k}^2}{d_{j,k}(2 - d_{j,k})}.
\tag{13}
$$

The theorem characterizes the steady-state variance of the frozen local recursion. We do not assume that the nonstationary training trajectory reaches this steady state at every iteration, and we use it to compare the local damping induced by different $\alpha$ and $\gamma$. Remark 1 derives order-of-magnitude bounds that are more useful in practice, separating two regimes: modes where graph damping dominates (low curvature) and modes where Hessian damping dominates (high curvature). These bounds are the key tool for understanding how $\gamma$ shapes the radius. Outside $0 < d_{j,k} < 2$, the scalar recursion is not mean-square contractive; this can occur in negative-curvature directions when $\alpha |\lambda_k| \geq \gamma \mu_j$.

**Remark 1** (Modal scaling bounds). *Suppose the conditions of Theorem 1 hold and $d_{j,k} \leq d_{\max} < 2$. Then*

$$
\frac{\alpha^2 \sigma_{j,k}^2}{2 d_{j,k}} \leq \mathbb{E}[z_{j,k}^2] \leq \frac{\alpha^2 \sigma_{j,k}^2}{(2 - d_{\max}) d_{j,k}}.
\tag{14}
$$

*If a mode is mixing-dominated, meaning $|\alpha \lambda_k| \leq \rho \gamma \mu_j$ for a fixed $\rho \in [0, 1)$, then*

$$
\mathbb{E}[z_{j,k}^2] = \Theta\left( \frac{\alpha^2 \sigma_{j,k}^2}{\gamma \mu_j} \right).
\tag{15}
$$

*If $\lambda_k > 0$, then*

$$\mathbb{E}[z_{j,k}^2] = \Theta\left(\frac{\alpha^2 \sigma_{j,k}^2}{\gamma \mu_j + \alpha \lambda_k}\right), \tag{16}$$

*so positive high-curvature directions are damped by the Hessian term. For negative-curvature directions, the same formula holds whenever $0 < d_{j,k} < 2$, but the Hessian term reduces $d_{j,k}$ and can increase the variance.*

The remark shows how the noise-weighted contribution of mixing-dominated modes scales with $\alpha^2/\gamma$, while positive high-curvature modes receive additional Hessian damping. Corollary 1 makes the aggregate low-curvature contribution explicit.

**Corollary 1** (Conditional radius scaling from low-curvature modes). *Let $\mathcal{S}_{\mathrm{low}} = \{(j,k) : 1 \leq j \leq n - 1, |\alpha \lambda_k| \leq \rho \gamma \mu_j\}$ for some fixed $\rho \in [0,1)$, and let $\mathcal{S}_{\mathrm{high}}^+ = \{(j,k) : 1 \leq j \leq n - 1, \lambda_k \geq c > 0\}$ for a fixed curvature threshold $c$. The set $\mathcal{S}_{\mathrm{low}}$ is non-empty only when genuinely flat directions exist, i.e., directions with $|\lambda_k| \ll \gamma \mu_j/\alpha$. Define the active low-curvature modal weight*

$$M_{\mathrm{low}}(\alpha, \gamma) := \sum_{(j,k) \in \mathcal{S}_{\mathrm{low}}} \frac{\sigma_{j,k}^2}{\mu_j}. \tag{17}$$

*If $M_{\mathrm{low}}(\alpha, \gamma) > 0$ and the modes in $\mathcal{S}_{\mathrm{low}}$ are stable, then*

$$\sum_{(j,k) \in \mathcal{S}_{\mathrm{low}}} \mathbb{E}[z_{j,k}^2] = \Theta\left(\frac{\alpha^2}{\gamma} M_{\mathrm{low}}(\alpha, \gamma)\right). \tag{18}$$

*If $M_{\mathrm{low}}(\alpha, \gamma)$ is bounded above and below by positive constants, this is $\Theta(\alpha^2/\gamma)$. If $\sigma_{j,k}^2 \leq \sigma_+^2 < \infty$ on $\mathcal{S}_{\mathrm{high}}^+$ and the corresponding modes are stable, then*

$$\sum_{(j,k) \in \mathcal{S}_{\mathrm{high}}^+} \mathbb{E}[z_{j,k}^2] = O(\alpha). \tag{19}$$

*The low-curvature contribution dominates when $\alpha^2/\gamma = \Omega(\alpha)$, i.e. when $\gamma = O(\alpha)$. For $\gamma = \alpha^p$, this holds for all $p \geq 1$; the practically relevant regime $p \geq 2$ is analyzed in Corollary 2.*

An exact null direction belongs to $\mathcal{S}_{\mathrm{low}}$ for every $\alpha, \gamma > 0$, whereas a fixed nonzero eigenvalue may leave the active set as the threshold $\gamma \mu_j/\alpha$ decreases. Moreover, non-emptiness of $\mathcal{S}_{\mathrm{low}}$ alone is insufficient: the gradient noise must inject non-vanishing modal weight. Low-rank Hessian structure in neural networks motivates the presence of flat directions (Keskar et al., 2016; Singh et al., 2021; Song et al., 2024; Ben Arous et al., 2024), while the persistence of the noise-weighted quantity $M_{\mathrm{low}}$ is the explicit condition used by the corollary.

The radius scaling in Corollary 1 is stated for general $\gamma$. Across frozen-window models with successively smaller $\alpha$, the active-mode condition increasingly favors flatter directions. This is a local spectral prediction rather than a guarantee that the full time-varying trajectory tracks the corresponding steady-state second moments. Corollary 2 specializes to the schedule $\gamma = \alpha^p$ used in DSGD-AC and gives a local prediction for how the maintained low-curvature contribution depends on $p$.

**Corollary 2** (Conditional scaling in active modes under $\gamma = \alpha^p$). *Set $\gamma = \alpha^p$ up to a fixed constant factor, with $p \geq 2$. For mixing-dominated modes, the active condition $|\lambda_k| \leq \rho \mu_j \alpha^{p-1}$ tightens as $\alpha \to 0$, so the following applies only to modes that remain active. For such modes,*

$$\mathbb{E}[z_{j,k}^2] = \Theta\left(\alpha^{2-p} \sigma_{j,k}^2/\mu_j\right). \tag{20}$$

*For $p = 2$, the variance is of constant order in each active mode. For $p > 2$, it grows as $\alpha \to 0$; if unchecked, this can destabilize training, consistent with the performance degradation observed for $p > 4$ in Table 1.*

These results show the mechanism behind DSGD-AC. Reducing $\gamma$ preserves disagreement magnitude in active low-curvature modes, while the Hessian term $\alpha \lambda_k$ continues to suppress positive high-curvature modes. For

$\gamma = \alpha^p$, $p = 2$ maintains a constant-order low-curvature contribution when its aggregate noise-weighted modal mass persists, whereas $p > 2$ increases that contribution in the frozen linearized model. As $\alpha$ decreases, the active set favors flatter directions, so the maintained radius concentrates there rather than spreading across the spectrum.

The curvature exposure $Q_t$ introduced in Section 3.3 can be written in modal coordinates as

$$Q_t = \sum_{i=1}^{n} (\delta_i^{(t)})^\top H_f \delta_i^{(t)} / \lambda_1(H_f) = \sum_{k=1}^{d} \lambda_k \sum_{j=1}^{n-1} (z_{j,k}^{(t)})^2 / \lambda_1(H_f). \tag{21}$$

This decomposition separates the two effects identified empirically: the magnitude of the maintained disagreement, controlled by $\gamma$, and its distribution across Hessian eigen directions, controlled by the modal noise variances and curvature-dependent damping $d_{j,k}$. A method with weaker normalized Hessian alignment can still produce larger $Q_t$ if its radius is sufficiently larger — precisely what DSGD-AC achieves.

**Remark 2** (Source-noise anisotropy and Hessian alignment). *The modal formula in Remark 1 implicitly explains why the above-random alignment $A_t > B_t$ observed in Figure 2b should not be attributed to adaptive consensus. If the injected noise $\sigma_{j,k}^2$ were isotropic in the Hessian eigen basis, the curvature term in $d_{j,k} = \gamma\mu_j + \alpha\lambda_k$ would damp high-curvature modes more strongly, and alignment would not exceed the random baseline. The observed alignment is instead consistent with anisotropic source noise: SGD noise in deep networks is known to concentrate energy in sharp directions (Wu et al., 2022; Ziyin et al., 2022), and Figure 2c verifies this directly for late-training gradient noise. Worker disagreements in both DSGD and DSGD-AC are driven by this structured noise, and graph mixing and curvature-dependent damping determine how much of each injected mode survives.*

## 4 Numerical Experiments

We evaluate DSGD-AC using Wide ResNets (Zagoruyko & Komodakis, 2016) on CIFAR-10 and CIFAR-100 (Krizhevsky et al., 2009) across different worker counts and communication topologies. The main experiments follow the SGD setting considered in our analysis and examine parameter sensitivity, generalization, loss curvature, and runtime efficiency. We additionally compare against momentum tracking (Takezawa et al., 2022) and SAM (Foret et al., 2020), and report a preliminary experiment combining adaptive consensus with decentralized Adam on Transformer-big and WMT14. We follow the hyperparameter settings from the original papers and reproduce comparable baseline performance to ensure fair comparisons. Unless otherwise specified, the image-classification models are trained for 200 epochs. Their global batch size is 128 with 8 workers and is linearly scaled with the number of workers, and the mixed-precision training is enabled by default. Since the networks use batch normalization layers and the training was not explicitly done on the global average center, we perform a calibration pass over the training set before evaluation, following Defazio et al. (2024)[1], which yields more reliable test performance. We report mean ± standard deviation from 3 repeat runs, and shaded regions indicate 95% confidence intervals. All experiments extend the decentralized training framework of Wang et al. (2025),[2] which overlaps decentralized communication with computation to hide communication latency. The training and evaluation details can be found in Appendix A.3.

### 4.1 Ablation study and sensitivity analysis

**Start epoch $E_{\text{start}}$** We introduce $E_{\text{start}}$ as the epoch from which the adaptive consensus mechanism is activated. Before $E_{\text{start}}$, $\gamma$ is set to 1 and DSGD-AC reduces to standard DSGD. In setups with more workers or weaker connectivity, decentralization already induces substantial consensus errors in early training, and reducing $\gamma$ below 1 at this stage further weakens mixing, hindering loss optimization before the model has converged to a stable region. Activating too late, on the other hand, leaves too few epochs for the regularization to act. Table 2 shows this two-sided sensitivity: $E_{\text{start}} = 100$ gives the best test accuracy, with performance degrading at both extremes, and the training loss keeps decreasing as $E_{\text{start}}$ increases

---

[1] https://github.com/facebookresearch/schedule_free
[2] https://github.com/WangZesen/Decent-DP

| $p$ | Test Acc. (%) ↑ | Train Loss ↓ | Test Loss ↓ |
|---|---|---|---|
| 0 | 96.04 ± 0.21 | 0.0006 ± 0.0000 | 0.182 ± 0.007 |
| 1 | 96.36 ± 0.06 | 0.0007 ± 0.0000 | 0.160 ± 0.002 |
| 2 | 96.48 ± 0.13 | 0.0017 ± 0.0001 | 0.150 ± 0.002 |
| 3 | 96.55 ± 0.10 | 0.0163 ± 0.0005 | 0.122 ± 0.002 |
| 4 | 96.41 ± 0.04 | 0.0347 ± 0.0010 | 0.118 ± 0.001 |
| 5 | 96.19 ± 0.05 | 0.0475 ± 0.0013 | 0.121 ± 0.002 |

Table 1: Sensitivity analysis of parameter $p$ in the WRN28-10 on CIFAR-10 experiments with 16 workers, one-peer ring topology, and $E_{\text{start}} = 100$. The best metric values are underlined.

| $E_{\text{start}}$ | Test Acc. (%) ↑ | Train Loss ↓ | Test Loss ↓ |
|---|---|---|---|
| 10 | 96.16 ± 0.06 | 0.0474 ± 0.0011 | 0.121 ± 0.003 |
| 50 | 96.31 ± 0.12 | 0.0395 ± 0.0006 | 0.121 ± 0.003 |
| 75 | 96.33 ± 0.09 | 0.0290 ± 0.0005 | 0.119 ± 0.002 |
| 100 | 96.55 ± 0.10 | 0.0163 ± 0.0005 | 0.122 ± 0.002 |
| 150 | 96.49 ± 0.08 | 0.0021 ± 0.0003 | 0.150 ± 0.002 |
| 175 | 96.21 ± 0.12 | 0.0008 ± 0.0001 | 0.181 ± 0.003 |

Table 2: Sensitivity analysis of $E_{\text{start}}$ in the WRN28-10 on CIFAR-10 experiments with 16 workers, one-peer ring topology, and $p = 3$. The best metric values are underlined.

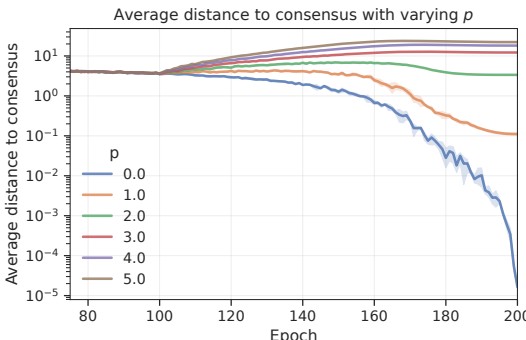

Figure 4: Average norm of consensus errors over epochs with varying $p$ in the WRN28-10 on CIFAR-10 experiments with 16 workers, the one-peer ring topology, and $E_{\text{start}} = 100$.

because of weaker regularization. As a practical heuristic, $E_{\text{start}}$ should follow the warm-up phase and scale with worker count and topology: more workers and weaker connectivity require a later activation.

**Parameter $p$**    Table 1 presents the sensitivity analysis with respect to $p$. The results indicate that the best test accuracy and test loss are achieved for $p \in [2, 4]$. Since DSGD-AC reduces to DSGD when $p = 0$, these experiments also serve as an ablation study of the adaptive consensus mechanism. With $p > 0$, DSGD-AC consistently achieves better performance than DSGD. Figure 4 presents the average norm of the consensus errors with various $p$. In line with the analysis in Section 3.4, the consensus errors persist for $p \geq 2$, and the regularization and generalization improvement become weak with a too-large $p$ because of worse alignment.

Appendix A.2.4 includes the training curves of experiments in the sensitivity analysis.

## 4.2  Tuned results with various setups

In Table 3, we summarize the experiment results of DSGD-AC with tuned $p$ and $E_{\text{start}}$ based on Tables 1 and 2, and the comparisons with SGD and DSGD in various setups. The algorithms are applied on training WRN28-10 and WRN16-8 on the CIFAR-100 dataset. The number of workers varies from 8 to 32, and the topologies are one-peer ring, one-peer exponential, and complete graph. According to the sensitivity analysis, we choose $p = 3$ for all DSGD-AC experiments, and $E_{\text{start}} =$10, 100, and 150 for 8-worker, 16-worker, and 32-worker setups, respectively. Since the best sharpness metric remains an open question, we report the top-1 Hessian eigenvalue as a surrogate, which has been empirically shown to correlate strongly with generalization (Bisla et al., 2022; Luo et al., 2024).

Across all evaluated setups, DSGD-AC achieves consistently higher test accuracy and lower test loss than both synchronous SGD and decentralized SGD (even under their optimal setups). These results validate the effectiveness of DSGD-AC across models, topologies, and system scales. The corresponding training curves

| Model | #W | Topo. | Algorithms (Test Accuracy (%) ↑ / Test Loss ↓ / Largest Eigenvalue ↓) | | |
|---|---|---|---|---|---|
| | | | Synchronous SGD | Decentralized SGD | DSGD-AC (ours) |
| 28-10 | 8 | ring | | $79.71_{\pm 0.13}$ / $1.108_{\pm 0.011}$ / $4.35_{\pm 1.19}$ | $\underline{82.12}_{\pm 0.12}$ / $\underline{0.897}_{\pm 0.007}$ / $0.59_{\pm 0.02}$ |
| | | exp | $80.00_{\pm 0.19}$ / $1.097_{\pm 0.014}$ / $4.53_{\pm 0.72}$ | $79.98_{\pm 0.33}$ / $1.100_{\pm 0.018}$ / $4.60_{\pm 1.20}$ | $81.82_{\pm 0.08}$ / $0.911_{\pm 0.013}$ / $0.53_{\pm 0.04}$ |
| | | complete | | $79.62_{\pm 0.05}$ / $1.114_{\pm 0.014}$ / $4.13_{\pm 0.85}$ | $81.41_{\pm 0.31}$ / $0.953_{\pm 0.001}$ / $\underline{0.47}_{\pm 0.09}$ |
| | 16 | ring | | $80.18_{\pm 0.14}$ / $1.050_{\pm 0.015}$ / $2.89_{\pm 1.01}$ | $\underline{82.19}_{\pm 0.17}$ / $\underline{0.846}_{\pm 0.012}$ / $0.51_{\pm 0.02}$ |
| | | exp | $79.79_{\pm 0.25}$ / $1.111_{\pm 0.021}$ / $6.75_{\pm 0.97}$ | $80.18_{\pm 0.28}$ / $1.066_{\pm 0.003}$ / $3.31_{\pm 0.82}$ | $82.09_{\pm 0.03}$ / $0.886_{\pm 0.003}$ / $0.57_{\pm 0.16}$ |
| | | complete | | $79.81_{\pm 0.29}$ / $1.094_{\pm 0.007}$ / $4.11_{\pm 0.40}$ | $81.64_{\pm 0.10}$ / $0.934_{\pm 0.010}$ / $\underline{0.45}_{\pm 0.03}$ |
| | 32 | ring | | $80.72_{\pm 0.34}$ / $0.950_{\pm 0.019}$ / $4.15_{\pm 0.61}$ | $\underline{81.92}_{\pm 0.13}$ / $\underline{0.831}_{\pm 0.006}$ / $0.62_{\pm 0.12}$ |
| | | exp | $79.90_{\pm 0.19}$ / $1.090_{\pm 0.019}$ / $4.02_{\pm 0.69}$ | $80.56_{\pm 0.21}$ / $1.000_{\pm 0.006}$ / $3.58_{\pm 1.06}$ | $\underline{81.92}_{\pm 0.04}$ / $0.886_{\pm 0.011}$ / $\underline{0.37}_{\pm 0.04}$ |
| | | complete | | $80.04_{\pm 0.12}$ / $1.060_{\pm 0.011}$ / $3.50_{\pm 1.09}$ | $81.58_{\pm 0.15}$ / $0.925_{\pm 0.001}$ / $0.42_{\pm 0.09}$ |
| 16-8 | 8 | ring | | $79.16_{\pm 0.36}$ / $0.986_{\pm 0.010}$ / $3.51_{\pm 0.60}$ | $\underline{80.54}_{\pm 0.31}$ / $0.910_{\pm 0.016}$ / $0.76_{\pm 0.03}$ |
| | | exp | $79.19_{\pm 0.13}$ / $0.997_{\pm 0.007}$ / $6.81_{\pm 2.34}$ | $78.90_{\pm 0.14}$ / $1.002_{\pm 0.012}$ / $4.71_{\pm 0.33}$ | $80.51_{\pm 0.08}$ / $\underline{0.902}_{\pm 0.004}$ / $\underline{0.75}_{\pm 0.04}$ |
| | | complete | | $79.12_{\pm 0.22}$ / $0.989_{\pm 0.012}$ / $8.89_{\pm 1.88}$ | $80.08_{\pm 0.20}$ / $0.931_{\pm 0.002}$ / $1.18_{\pm 0.76}$ |
| | 16 | ring | | $79.07_{\pm 0.22}$ / $0.979_{\pm 0.008}$ / $4.01_{\pm 1.36}$ | $\underline{80.51}_{\pm 0.11}$ / $\underline{0.861}_{\pm 0.015}$ / $0.56_{\pm 0.01}$ |
| | | exp | $79.02_{\pm 0.27}$ / $1.012_{\pm 0.014}$ / $6.32_{\pm 1.21}$ | $79.02_{\pm 0.14}$ / $0.982_{\pm 0.011}$ / $4.91_{\pm 0.37}$ | $80.44_{\pm 0.08}$ / $0.889_{\pm 0.006}$ / $0.57_{\pm 0.02}$ |
| | | complete | | $79.03_{\pm 0.22}$ / $1.001_{\pm 0.012}$ / $5.32_{\pm 2.60}$ | $80.01_{\pm 0.06}$ / $0.919_{\pm 0.016}$ / $\underline{0.51}_{\pm 0.01}$ |
| | 32 | ring | | $79.12_{\pm 0.24}$ / $0.976_{\pm 0.006}$ / $2.95_{\pm 0.39}$ | $80.17_{\pm 0.10}$ / $\underline{0.872}_{\pm 0.005}$ / $0.72_{\pm 0.21}$ |
| | | exp | $78.90_{\pm 0.14}$ / $1.010_{\pm 0.009}$ / $8.41_{\pm 1.13}$ | $79.13_{\pm 0.04}$ / $0.974_{\pm 0.004}$ / $3.17_{\pm 0.19}$ | $\underline{80.22}_{\pm 0.16}$ / $0.921_{\pm 0.009}$ / $\underline{0.51}_{\pm 0.03}$ |
| | | complete | | $79.04_{\pm 0.19}$ / $0.988_{\pm 0.014}$ / $2.81_{\pm 0.36}$ | $79.73_{\pm 0.13}$ / $0.955_{\pm 0.006}$ / $0.52_{\pm 0.01}$ |

Table 3: Performance comparison of synchronous SGD, decentralized SGD, and DSGD-AC. Dataset: CIFAR-100. Model: WRN28-10 and WRN16-8. Number of workers: 8, 16, 32. Topology: one-peer ring, one-peer exponential graph (Ying et al., 2021), and complete graph. $p = 3$ for all DSGD-AC experiments. $E_{\text{start}}$ is set to 10, 100 and 150 for 8-worker, 16-worker and 32-worker setups, respectively. The best metric values for each distributed setup are underlined.

can be found in Appendix A.2.3. DSGD-AC also consistently reaches solutions with substantially smaller dominant Hessian eigenvalues than standard DSGD and synchronous SGD.

## 4.3 Complementary comparisons and extensions

We complement the main experiments with three additional comparisons. First, underlined momentum tracking (Takezawa et al., 2022), a gradient-tracking method (Nedic et al., 2017) that actively promotes consensus, remains close to standard SGD and DSGD in our i.i.d. setting and underperforms DSGD-AC across both models and all three topologies. This matches the benefit of preserving controlled late-training disagreement (Appendix A.2.6). Second, SAM (Foret et al., 2020) provides a higher-cost sharpness-aware reference: at 200 epochs, it achieves higher accuracy and lower test loss than DSGD-AC but requires a second gradient evaluation and approximately $4.9\times$ its wall-clock time; at the same FLOP budget, 100-epoch SAM achieves comparable accuracy to DSGD-AC but remains $\sim 2.4\times$ slower (Appendix A.2.5). Finally, although our analysis focuses on unpreconditioned DSGD, applying learning-rate-synchronized consensus scaling to decentralized Adam on Transformer-big (Vaswani et al., 2017) trained on WMT14 (Bojar et al., 2014) improves test metrics over other baselines. Since these gains are modest and adaptive preconditioning changes the gradient-noise geometry underlying our analysis, we treat the DAdam-AC result as preliminary evidence of compatibility rather than validation of an optimizer-independent mechanism (Appendix A.2.2).

## 4.4 Training efficiency

Table 4 shows the training time with WRN28-10 and CIFAR-100 using 1, 2, and 4 8×T4 nodes interconnected by 100Gbps InfiniBand. Thanks to the decentralized communication pattern, decentralized training can hide the communication time and provide more resilience to stragglers, which makes it more efficient than synchronous training even under the single-node setup (Lian et al., 2017; Assran et al., 2019; Wang et al., 2025). In these experiments, decentralized methods require only 0.78–0.85× the training time of synchronous

| # Nodes / Workers | SGD | DSGD | DSGD-AC |
|---|---|---|---|
| 1 / 8 | $91.50 \pm 0.21$ ($\times 1.000$) | $77.69 \pm 0.45$ ($\times 0.849$) | $78.11 \pm 0.52$ ($\times 0.854$) |
| 2 / 16 | $48.38 \pm 0.11$ ($\times 1.000$) | $39.09 \pm 0.19$ ($\times 0.808$) | $39.69 \pm 0.14$ ($\times 0.820$) |
| 4 / 32 | $25.78 \pm 0.07$ ($\times 1.000$) | $20.08 \pm 0.12$ ($\times 0.779$) | $20.28 \pm 0.14$ ($\times 0.787$) |

Table 4: Training time (in minutes) of SGD, DSGD, and DSGD-AC in the WRN28-10 on CIFAR-100 experiments. The communication topology for decentralized methods is the one-peer ring topology. $\times x$ is the relative time compared with synchronous SGD in the same distributed setup.

SGD, and the relative speedup further improves as the number of workers increases. Compared with DSGD, DSGD-AC introduces negligible additional runtime overhead.

## 5 Conclusion and discussion

Prior work on decentralized training has focused on reducing consensus errors, viewing them as an obstacle to convergence and generalization. Our analysis and experiments support an alternative view: controlled disagreement can serve as an implicit regularizer. Under a frozen local linearization, the modal analysis characterizes how $\gamma^{(t)}$ balances disagreement radius and curvature profile and thereby preserves curvature exposure, which is empirically associated with flatter solutions. Across the evaluated settings, DSGD-AC achieves better test performance than standard DSGD and synchronous SGD at negligible additional cost. The key theoretical contribution is a modal analysis of the consensus-error dynamics, which characterizes how the adaptive factor $\gamma^{(t)}$ can balance radius and curvature profile through the steady-state variances of graph-Hessian modes. This analysis explains how DSGD-AC can preserve curvature exposure under a frozen local linearization, while the experiments show that it does so better than standard DSGD in the evaluated settings. Together with the scalability inherited from decentralized training, it positions DSGD-AC as a compelling alternative to synchronous SGD for distributed training.

**Limitation and future work** Several limitations merit attention. First, DSGD-AC achieves a weaker normalized Hessian alignment than standard DSGD, as shown in Figure 2. The analysis in Section 3.3 shows this does not undermine the curvature exposure argument, since the larger radius compensates. However, it suggests that the linear mixing operator with a decaying scalar factor is not optimal: a mixing operator designed to preserve alignment while maintaining radius could improve the regularization effect further. Second, the mechanism relies on the anisotropy of SGD gradient noise, which aligns naturally with the Hessian eigenspace. Adaptive optimizers such as Adam (Kingma & Ba, 2014) modulate this noise, disrupting the anisotropy and weakening the alignment of consensus errors with sharp directions. Extending adaptive consensus to such optimizers is therefore non-trivial and remains an open problem. Whether DSGD-AC finds different basins of attraction than DSGD, and what role escape dynamics play in the observed generalization improvement, are further open questions. Finally, our modal theory is a local frozen-window analysis. As discussed in Appendix A.1.7, standard DSGD bounds suggest that bounded maintained disagreement corresponds to convergence to a wider neighborhood, whereas an unbounded disagreement radius could make the bound vacuous and permit divergence. The evaluated finite-horizon runs maintain bounded consensus-error norms, but establishing global convergence for the full time-varying DSGD-AC trajectory remains future work.

**Broader Impact Statement**

The proposed method may contribute to more efficient distributed training by improving the generalization performance of decentralized optimization without requiring extra gradient evaluations or substantial memory overhead. Such efficiency gains could reduce communication costs and, in some settings, lower the computational resources needed to reach a target accuracy. However, efficiency improvements may also encourage larger-scale training runs, potentially offsetting environmental benefits. We therefore view DSGD-AC as a tool for improving the quality-efficiency trade-off, rather than as a standalone solution to the environmental costs of modern deep learning.

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

# A  Appendix

## A.1  Proofs and derivations

### A.1.1  Derivation of Eq. (5)

We detail the algebra behind the surrogate view in Eq. (5). First, the objective-at-the-deployed-model term and the sharpness term satisfy

$$\sum_{i=1}^{n} f_i(\bar{x}) + \sum_{i=1}^{n} [f_i(x_i) - f_i(\bar{x})] = \sum_{i=1}^{n} f_i(x_i). \tag{22}$$

Consequently, their derivative with respect to each $x_i$ is simply the local loss gradient $\nabla f_i(x_i)$. In matrix form, the derivative is the stack of these local gradients. Thus, a stochastic gradient step on these two terms gives the local-objective part of the DSGD update. It remains to verify that the final term gives the mixing step. Consider only the quadratic consensus penalty

$$\mathcal{R}^{(t)}(x_1, \ldots, x_n) = \frac{1}{4\alpha^{(t)}} \sum_{i,j \in [n]} W_{ij} \|x_i - x_j\|^2. \tag{23}$$

Since $W$ is symmetric and doubly stochastic, $L = I - W$ is symmetric and

$$\sum_{i,j \in [n]} W_{ij} \|x_i - x_j\|^2 = 2 \operatorname{tr}(X^\top L X), \tag{24}$$

where $X = [x_1, \ldots, x_n]^\top$. Hence

$$\nabla_X \mathcal{R}^{(t)}(X) = \frac{1}{\alpha^{(t)}} L X. \tag{25}$$

A gradient step of size $\alpha^{(t)}$ on this penalty gives

$$X - \alpha^{(t)} \nabla_X \mathcal{R}^{(t)}(X) = X - LX = WX = X + (W - I)X, \tag{26}$$

which is exactly the DSGD mixing step. Combining this penalty step with the stochastic gradient step on the local objectives gives the matrix-form DSGD update in Eq. (4). For DSGD-AC, replacing $\mathcal{R}^{(t)}$ by $\gamma^{(t)} \mathcal{R}^{(t)}$ yields the scaled mixing step $\gamma^{(t)}(W - I)X$.

### A.1.2 Proof of Lemma 1

*Proof.* Since $P\mathbf{1} = 0$ and $P\Delta^{(t-1)} = \Delta^{(t-1)}$ (as $\sum_i \delta_i^{(t-1)} = 0$ by the definition of $\Delta$), applying Assumption 4 to the disagreement recursion Equation 11 and invoking Assumption 5 ($H^{(t)} = H_f$, $R^{(t)} = 0$, $\alpha^{(t)} = \alpha$, and $\gamma^{(t)} = \gamma$) gives

$$
\begin{aligned}
\Delta^{(t)} &= (I - \gamma L)\Delta^{(t-1)} - \alpha P G^{(t)} \\
&= (I - \gamma L)\Delta^{(t-1)} - \alpha P(\mathbf{1}\nabla f(\bar{x}^{(t-1)})^\top + \Delta^{(t-1)} H^{(t)} + R^{(t)} + \Xi^{(t)}) \\
&= (I - \gamma L)\Delta^{(t-1)} - \alpha P(\Delta^{(t-1)} H^{(t)} + R^{(t)} + \Xi^{(t)}) \\
&= (I - \gamma L)\Delta^{(t-1)} - \alpha \Delta^{(t-1)} H_f - \alpha P \Xi^{(t)}.
\end{aligned}
\tag{27}
$$

Multiplying Eq. (27) by $u_k$ on the right and using $H_f u_k = \lambda_k u_k$ gives

$$\Delta^{(t)} u_k = (I - \gamma L - \alpha \lambda_k I)\Delta^{(t-1)} u_k - \alpha P \Xi^{(t)} u_k. \tag{28}$$

For every disagreement mode $j \in \{1, \ldots, n-1\}$, $Pv_j = v_j$ and $Lv_j = \mu_j v_j$. Multiplying by $v_j^\top$ on the left and setting $z_{j,k}^{(t)} := v_j^\top \Delta^{(t)} u_k$, $\xi_{j,k}^{(t)} := v_j^\top \Xi^{(t)} u_k$ yields the claimed recursion. $\square$

### A.1.3 Proof of Theorem 1

*Proof.* Lemma 1 gives

$$z_{j,k}^{(t)} = a_{j,k} z_{j,k}^{(t-1)} - \alpha \xi_{j,k}^{(t)}, \quad a_{j,k} := 1 - d_{j,k}. \tag{29}$$

Since $z_{j,k}^{(t-1)}$ is $\mathcal{F}_{t-1}$-measurable and $\mathbb{E}[\xi_{j,k}^{(t)} \mid \mathcal{F}_{t-1}] = 0$, squaring and taking conditional expectation yields

$$\mathbb{E}[(z_{j,k}^{(t)})^2 \mid \mathcal{F}_{t-1}] = a_{j,k}^2 (z_{j,k}^{(t-1)})^2 + \alpha^2 \sigma_{j,k}^2. \tag{30}$$

Taking total expectation and writing $V_t := \mathbb{E}[(z_{j,k}^{(t)})^2]$ gives

$$V_t = a_{j,k}^2 V_{t-1} + \alpha^2 \sigma_{j,k}^2. \tag{31}$$

The condition $0 < d_{j,k} < 2$ is equivalent to $|a_{j,k}| < 1$, so this second-moment recursion is contractive and converges from any finite $V_0$ to its unique fixed point $V$. This fixed point satisfies

$$V = a_{j,k}^2 V + \alpha^2 \sigma_{j,k}^2. \tag{32}$$

Since $|a_{j,k}| < 1$ implies $1 - a_{j,k}^2 \neq 0$, this equation has the unique solution

$$V = \frac{\alpha^2 \sigma_{j,k}^2}{1 - a_{j,k}^2} = \frac{\alpha^2 \sigma_{j,k}^2}{d_{j,k}(2 - d_{j,k})}, \tag{33}$$

where the last equality uses $1 - a_{j,k}^2 = (1 - a_{j,k})(1 + a_{j,k}) = d_{j,k}(2 - d_{j,k})$. $\square$

### A.1.4 Derivation for Remark 1

*Derivation.* Since $0 < d_{j,k} \leq d_{\max} < 2$, the bounds $2 - d_{\max} \leq 2 - d_{j,k} \leq 2$ applied to Theorem 1 give

$$\frac{\alpha^2 \sigma_{j,k}^2}{2d_{j,k}} \leq \mathbb{E}[z_{j,k}^2] \leq \frac{\alpha^2 \sigma_{j,k}^2}{(2 - d_{\max})d_{j,k}}. \tag{34}$$

For mixing-dominated modes, $|\alpha\lambda_k| \leq \rho\gamma\mu_j$ implies

$$(1 - \rho)\gamma\mu_j \leq d_{j,k} \leq (1 + \rho)\gamma\mu_j. \tag{35}$$

Together with $d_{j,k} \leq d_{\max} < 2$, this makes $d_{j,k}(2 - d_{j,k}) = \Theta(\gamma\mu_j)$. Substituting into Eq. (13) proves the low-curvature claim. For $\lambda_k > 0$, the denominator is

$$d_{j,k}(2 - d_{j,k}) = \Theta(d_{j,k}) = \Theta(\gamma\mu_j + \alpha\lambda_k), \tag{36}$$

where the constants depend on $d_{\max}$. The same identity remains true for negative $\lambda_k$ whenever $0 < d_{j,k} < 2$, but negative curvature decreases $d_{j,k}$ relative to $\gamma\mu_j$. □

### A.1.5 Proof of Corollary 1

*Proof.* By Parseval's identity for the orthonormal bases $V$ and $U$,

$$\|\Delta\|_F^2 = \sum_{j=1}^{n-1} \sum_{k=1}^{d} z_{j,k}^2. \tag{37}$$

Remark 1 gives

$$\mathbb{E}[z_{j,k}^2] = \Theta\left(\frac{\alpha^2 \sigma_{j,k}^2}{\gamma\mu_j}\right) \tag{38}$$

for every $(j,k) \in \mathcal{S}_{\text{low}}$. Summing over the active low-curvature modes yields

$$\sum_{(j,k) \in \mathcal{S}_{\text{low}}} \mathbb{E}[z_{j,k}^2] = \Theta\left(\frac{\alpha^2}{\gamma} \sum_{(j,k) \in \mathcal{S}_{\text{low}}} \frac{\sigma_{j,k}^2}{\mu_j}\right) = \Theta\left(\frac{\alpha^2}{\gamma} M_{\text{low}}(\alpha, \gamma)\right). \tag{39}$$

If $M_{\text{low}}(\alpha, \gamma)$ is bounded above and below by positive constants in the local window, this reduces to $\Theta(\alpha^2/\gamma)$. For every $(j,k) \in \mathcal{S}_{\text{high}}^+$, $\lambda_k \geq c > 0$, so

$$d_{j,k} = \gamma\mu_j + \alpha\lambda_k \geq \alpha c. \tag{40}$$

Since $d_{j,k} \leq d_{\max} < 2$, Eq. (13) gives

$$\mathbb{E}[z_{j,k}^2] \leq C \frac{\alpha^2 \sigma_{j,k}^2}{\alpha c} = O(\alpha) \tag{41}$$

for a constant $C$ independent of $\alpha$. Summing over the finite set $\mathcal{S}_{\text{high}}^+$ preserves the $O(\alpha)$ rate. If $M_{\text{low}}(\alpha, \gamma)$ remains non-negligible and $\alpha^2/\gamma = \Omega(\alpha)$, the low-curvature contribution decays no faster than this high-curvature upper bound. □

### A.1.6 Proof of Corollary 2

*Proof.* If $\gamma = \alpha^p$ up to a constant factor, the mixing-dominated condition

$$|\alpha\lambda_k| \leq \rho\gamma\mu_j \tag{42}$$

is equivalent to $|\lambda_k| \leq C\rho\mu_j\alpha^{p-1}$ for a fixed constant $C > 0$. For such modes, Remark 1 gives

$$\mathbb{E}[z_{j,k}^2] = \Theta\left(\frac{\alpha^2 \sigma_{j,k}^2}{\alpha^p \mu_j}\right) = \Theta\left(\alpha^{2-p} \sigma_{j,k}^2/\mu_j\right). \tag{43}$$

For $p = 2$ this is constant order in $\alpha$ for each active mixing-dominated mode; for $p > 2$ it grows in the linear stationary approximation for each such mode. The condition $|\lambda_k| \leq O(\alpha^{p-1})$ shrinks as $\alpha$ decreases, so the statement applies only to modes that remain active in this mixing-dominated regime. □

### A.1.7 Discussion: convergence implications of maintained disagreement

Standard DSGD analyses bound the stationarity of the average model. The bound has three parts: an optimization term, a stochastic-noise term, and a term controlled by the consensus errors (Lian et al., 2017; Koloskova et al., 2020; Alghunaim & Yuan, 2022). For example, the non-convex descent lemma of Koloskova et al. (2020, Appendix C.2, Lemma 10), specialized to the i.i.d. and uniformly bounded-variance setting, gives the following representative finite-horizon bound under the additional standard assumptions that $f$ is $L$-smooth and a constant step size $\alpha < 1/(4L)$ is used:

$$\frac{1}{T}\sum_{t=1}^{T}\mathbb{E}\Big[\|\nabla f(\bar{x}^{(t)})\|^2\Big] \leq \underbrace{\frac{4\big(f(\bar{x}^{(1)}) - f_{\inf}\big)}{\alpha T} + \frac{4L\alpha\widehat{\sigma}^2}{n}}_{\mathcal{B}_{\mathrm{opt/noise}}(T)} + \underbrace{\frac{4L^2}{n}}_{C_\Delta}\frac{1}{T}\sum_{t=1}^{T}\mathbb{E}\Big[\|\Delta^{(t)}\|_F^2\Big], \tag{44}$$

Here $T$ is the number of iterations, $f_{\inf} := \inf_x f(x) > -\infty$, and the expectation is over the sampled stochastic gradients. The quantity $\widehat{\sigma}^2$ bounds their aggregate variance, in the sense that $\mathbb{E}\|n^{-1}\sum_{i=1}^{n}\xi_i^{(t)}\|^2 \leq \widehat{\sigma}^2/n$, while $n$, $\bar{x}^{(t)}$, and $\Delta^{(t)}$ are the number of workers, their average model, and their disagreement matrix defined in the problem-setting section. The correspondence with our notation follows from $\|\Delta^{(t)}\|_F^2 = \sum_{i=1}^{n}\|x_i^{(t)} - \bar{x}^{(t)}\|^2$. Thus, $\mathcal{B}_{\mathrm{opt/noise}}(T)$ collects the initial-suboptimality and stochastic-noise contributions, while the non-negative coefficient $C_\Delta = 4L^2/n$ measures how disagreement enters this representative smoothness-based bound. Other analyses and step-size schedules change these constants and the precise optimization/noise terms, but retain the same qualitative dependence on the average consensus error.

Under a decaying step-size schedule for which standard DSGD has vanishing consensus errors, the second term disappears asymptotically. If DSGD-AC instead maintains a uniformly bounded radius $\|\Delta^{(t)}\|_F \leq R$, this contribution remains bounded by $C_\Delta R^2$, and, when the remaining terms vanish, the method can be interpreted as converging to a wider stationarity neighborhood. If the radius grows without bound, however, the bound becomes vacuous and divergence is possible.

However, it should be noted that the training horizon is usually finite in practice, and Figures 1b and 4 show finite consensus-error norms throughout the evaluated runs. The sensitivity analysis further shows that overly aggressive disagreement preservation increases training loss, whereas the selected values of $p$ remain stable and improve test performance. This behavior is consistent with the intended optimization–generalization trade-off: our objective is improved generalization of the deployed average model rather than exact consensus or minimum training loss. These observations provide finite-horizon evidence, not a global asymptotic convergence guarantee.

## A.2 Additional experiments and details

### A.2.1 Empirical scope of the quadratic loss-envelope diagnostic

The curvature-exposure diagnostic in Eq. (6) uses a second-order expansion around the deployed average model. Because DSGD-AC deliberately maintains a larger disagreement radius than DSGD, higher-order terms need not be negligible. We therefore evaluate how much of the exact loss envelope is captured by its quadratic component along the worker deviations observed during training in the DSGD-AC running example.

Let

$$\phi(\bar{x}, \Delta) := \sum_{i=1}^{n} f(\bar{x} + \delta_i) \tag{45}$$

denote the exact aggregate loss at the worker models. At checkpoint $t$, define

$$
\begin{aligned}
C_0^{(t)} &:= nf(\bar{x}^{(t)}), \\
C_1^{(t)} &:= \sum_{i=1}^{n} \nabla f(\bar{x}^{(t)})^{\top} \delta_i^{(t)} = 0, \\
C_2^{(t)} &:= \frac{1}{2} \sum_{i=1}^{n} (\delta_i^{(t)})^{\top} H^{(t)} \delta_i^{(t)}, \\
C_{\text{odd}}^{(t)} &:= \frac{1}{2} \left[ \phi(\bar{x}^{(t)}, \Delta^{(t)}) - \phi(\bar{x}^{(t)}, -\Delta^{(t)}) \right], \\
C_{\text{even}}^{(t)} &:= \frac{1}{2} \left[ \phi(\bar{x}^{(t)}, \Delta^{(t)}) + \phi(\bar{x}^{(t)}, -\Delta^{(t)}) \right] - C_0^{(t)}, \\
R_{\text{even}}^{(t)} &:= C_{\text{even}}^{(t)} - C_2^{(t)}, \\
R_{\text{total}}^{(t)} &:= \phi(\bar{x}^{(t)}, \Delta^{(t)}) - C_0^{(t)} - C_2^{(t)} = C_{\text{odd}}^{(t)} + R_{\text{even}}^{(t)}.
\end{aligned}
\tag{46}
$$

The first-order term vanishes exactly because $\sum_i \delta_i^{(t)} = 0$. The symmetry-based quantities $C_{\text{odd}}^{(t)}$ and $C_{\text{even}}^{(t)}$ are exact: the former collects all odd-order contributions, whereas the latter collects all non-constant even-order contributions. Consequently, $R_{\text{even}}^{(t)}$ contains the fourth- and higher-order even contributions, and $R_{\text{total}}^{(t)}$ is the total error incurred by retaining only the quadratic term.

We report the residual relative to the exact aggregate loss,

$$
\eta_t := \frac{|R_{\text{total}}^{(t)}|}{|\phi(\bar{x}^{(t)}, \Delta^{(t)})|}.
\tag{47}
$$

This quantity directly measures the relative error of the approximation $\phi(\bar{x}^{(t)}, \Delta^{(t)}) \approx C_0^{(t)} + C_2^{(t)}$. All losses and Hessian-vector products are evaluated on the full 50,000-example training set using the same model state, an evaluation batch size of 16, and a fixed data seed at each saved checkpoint.

| $E$ | $\phi$ | $C_0$ | $C_2$ | $C_{\text{odd}}$ | $R_{\text{even}}$ | $R_{\text{total}}$ | $|R_{\text{total}}|/|\phi|$ |
|---|---|---|---|---|---|---|---|
| 100 | 5.4088 | 2.6789 | 2.0045 | 0.1970 | 0.5285 | 0.7254 | 0.134 |
| 110 | 5.8844 | 2.2299 | 2.3188 | 0.3595 | 0.9761 | 1.3357 | 0.227 |
| 120 | 5.1489 | 1.8300 | 1.9963 | 0.3862 | 0.9364 | 1.3225 | 0.257 |
| 130 | 5.1498 | 1.4641 | 1.9983 | 0.4878 | 1.1996 | 1.6874 | 0.328 |
| 140 | 4.5375 | 1.1235 | 1.6931 | 0.5078 | 1.2130 | 1.7208 | 0.379 |
| 150 | 3.7562 | 0.7754 | 1.3342 | 0.4831 | 1.1635 | 1.6466 | 0.438 |
| 160 | 2.4431 | 0.4684 | 0.8449 | 0.3084 | 0.8214 | 1.1298 | 0.462 |
| 170 | 1.2374 | 0.2607 | 0.4443 | 0.1305 | 0.4019 | 0.5324 | 0.430 |
| 180 | 0.5362 | 0.1597 | 0.2109 | 0.0336 | 0.1320 | 0.1656 | 0.309 |
| 190 | 0.3313 | 0.1311 | 0.1308 | 0.0090 | 0.0604 | 0.0694 | 0.210 |
| 200 | 0.3013 | 0.1272 | 0.1191 | 0.0048 | 0.0502 | 0.0550 | 0.183 |

Table 5: Decomposition of the exact loss envelope along the worker deviations observed from epochs 100 to 200 when training WRN16-8 on CIFAR-100 with DSGD-AC. The final column reports the error of the quadratic approximation relative to the exact aggregate loss. Values may differ slightly from identities computed from the displayed entries because of rounding.

Table 5 shows that the accuracy of the quadratic approximation is non-monotone over the evaluation window. Its relative residual increases from 13.4% at epoch 100 to 46.2% at epoch 160, before decreasing to 18.3%

at epoch 200. The approximation is therefore least accurate in the middle of this window and becomes substantially more accurate as training approaches its final solution. The aggregate odd-order contribution peaks near epoch 140 and subsequently decreases from 0.5078 to 0.0048, amounting to only 1.6% of the exact aggregate loss at the final checkpoint. The remaining final-checkpoint error is dominated by fourth- and higher-order even terms.

Although higher-order terms are non-negligible and vary over training, the final-stage decrease in the residual implies that $C_0 + C_2$ provides an increasingly accurate description of the aggregate worker loss near the final solution, which is the regime targeted by our local analysis. We therefore use the exactly computed $C_2$ as a measure of local curvature exposure, while avoiding the claim that it is an exact surrogate for the complete finite-radius loss envelope. The exact perturbation-ensemble interpretation does not require higher-order terms to vanish. The modal analysis is separate and relies on the local gradient linearization in Assumption 4.

### A.2.2 Decentralized Adam with adaptive consensus

We also evaluate the compatibility of the idea of controlling consensus errors on transformer models by simply replacing the local update with the Adam optimizer (Kingma & Ba, 2014). DSGD-AC is then adapted to DAdam-AC as in Algorithm 2.

---

**Algorithm 2:** Decentralized Adam with adaptive consensus (DAdam-AC) on worker $i$

---

**Data:** Dataset $(D)$, the number of workers $(N)$, the number of epochs $(E)$, the number of batches per epoch $(T)$, initialization $(x^{(0)})$, and a hyperparameter $(p \in \mathbb{R}^+)$.

**Result:** Deployed model $\bar{x} = \frac{1}{n} \sum_{j=1}^{n} x_j^{(TE)}$

$x_1^{(0)} = x_2^{(0)} = \cdots = x_n^{(0)} = x^{(0)}$

**for** $t = 1$ **to** $TE$ **do**

$\quad g_i^{(t)} = \nabla f(x_i^{(t-1)}; s_i^{(t)})$

$\quad m_i^{(t)} = \beta_1 m_i^{(t-1)} + (1 - \beta_1) g_i^{(t)}$

$\quad v_i^{(t)} = \beta_2 v_i^{(t-1)} + (1 - \beta_2) g_i^{(t)} \odot g_i^{(t)}$

$\quad \hat{m}_i^{(t)} = m_i^{(t)} / (1 - \beta_1^t)$

$\quad \hat{v}_i^{(t)} = v_i^{(t)} / (1 - \beta_2^t)$

$\quad \gamma^{(t)} = \left[ \alpha^{(t)} / \alpha_{\max} \right]^p$

$\quad x_i^{(t)} = x_i^{(t-1)} - \alpha^{(t)} \hat{m}_i^{(t)} / (\sqrt{\hat{v}_i^{(t)}} + \epsilon) + \gamma^{(t)} \sum_{j \in \mathcal{N}(i)} W_{ij}(x_j^{(t-1)} - x_i^{(t-1)})$

**end**

---

We train Transformer (the big variant, $\sim$213M parameters) (Vaswani et al., 2017) on WMT14 (English-to-German) (Bojar et al., 2014) and present the curves of training losses and BLEU scores on the test set. The BLEU scores (Papineni et al., 2002) (which are used to evaluate the translation quality in the original paper) and the losses on the test set and the training set are reported in Table 6.

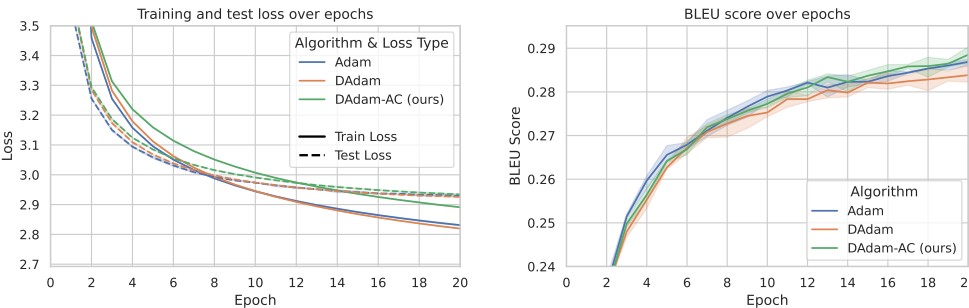

Figure 5: Transformer (big) on WMT14 English-to-German. **Left**: Losses on training set. **Right**: BLEU scores on the test set.

| Algorithm | BLEU score ↑ | Test loss ↓ | Train loss ↓ |
|-----------|--------------|-------------|--------------|
| Adam | $28.68 \pm 0.07$ | $2.9290 \pm 0.0026$ | $2.8310 \pm 0.0019$ |
| DAdam | $28.38 \pm 0.22$ | $2.9258 \pm 0.0018$ | $\underline{2.8195 \pm 0.0008}$ |
| DAdam-AC | $\underline{28.89 \pm 0.17}$ | $\underline{2.9205 \pm 0.0020}$ | $2.8456 \pm 0.0016$ |

Table 6: Performance comparison of DAdam, Adam, and DAdam-AC on neural machine translation with the transformer model.

DAdam-AC obtains the best mean BLEU score and test loss among the three methods. We therefore treat this as preliminary evidence of compatibility rather than validation of the DSGD-AC mechanism for adaptive optimizers.

### A.2.3 Training curves

Figures 6 to 11 present the training curves corresponding to the experiments in Table 3. The figures show the impact of the adaptive consensus after the epoch it is activated, and the trend that DSGD-AC trades training loss for better test loss and test accuracy. Moreover, from the figures, one can observe that there is an increasing trend in test loss in the last 30-50 epochs of training. Based on the analysis in Section 3.4, a heuristic explanation is that the regularization becomes weaker on the top eigenvalues due to the worse alignment at the late-training phase, which hinders the test performance.

In the training curve plots, the choice of topology causes more deviation when training with DSGD-AC, while DSGD-AC still outperforms DSGD consistently across various topologies. The modal analysis in Eq. (13) explains the phenomenon. DSGD-AC decreases $\gamma^{(t)}$ with the learning rate to preserve disagreement and, therefore, retains an explicit dependence on the graph spectrum. This changes the strength of the implicit regularization and explains why DSGD-AC shows a somewhat larger spread across topologies.

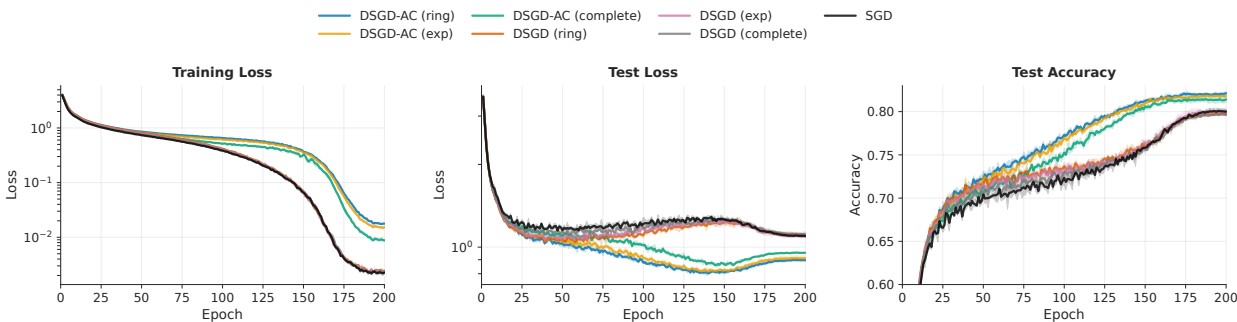

Figure 6: WRN28-10, CIFAR-100, 8 workers

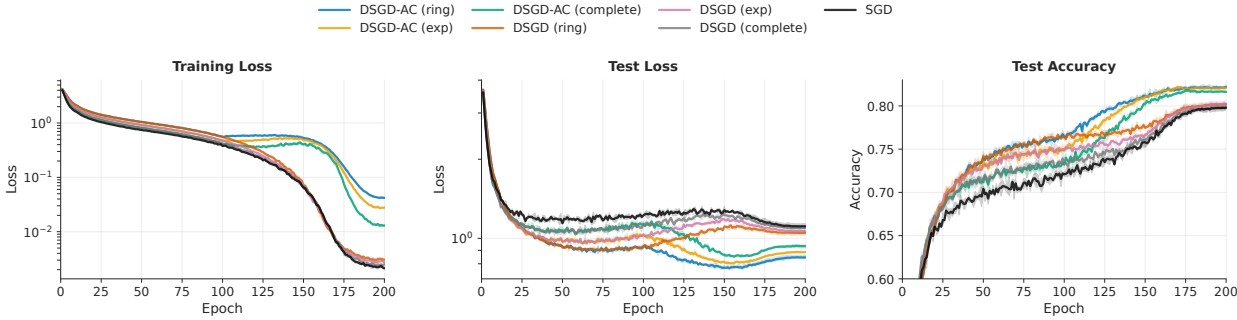

Figure 7: WRN28-10, CIFAR-100, 16 workers

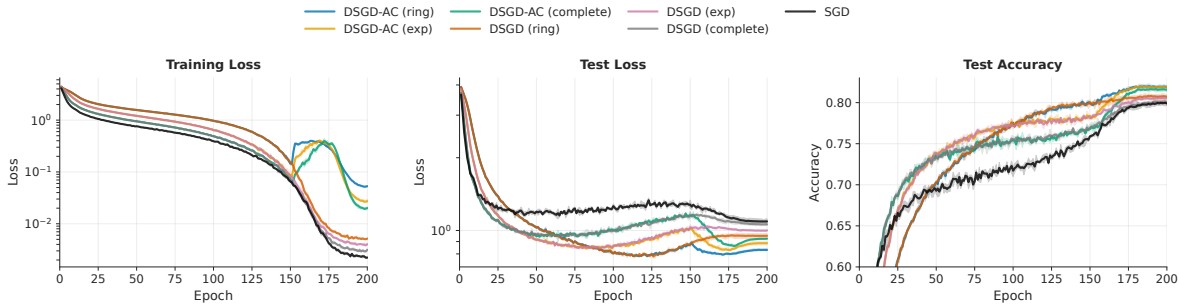

Figure 8: WRN28-10, CIFAR-100, 32 workers

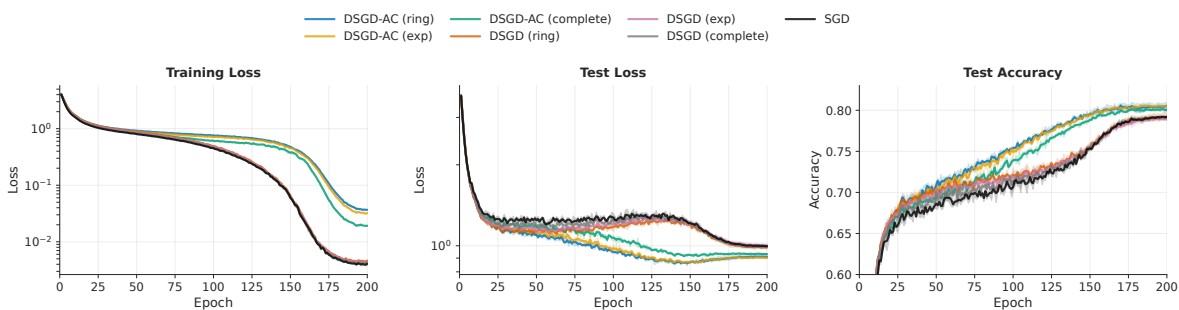

Figure 9: WRN16-8, CIFAR-100, 8 workers

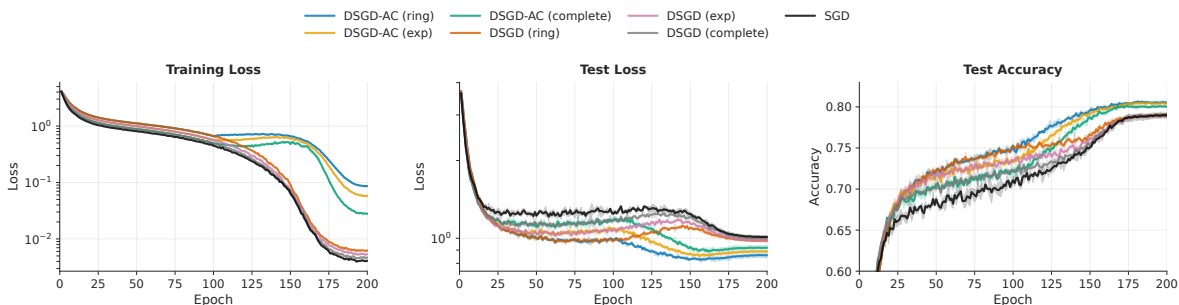

Figure 10: WRN16-8, CIFAR-100, 16 workers

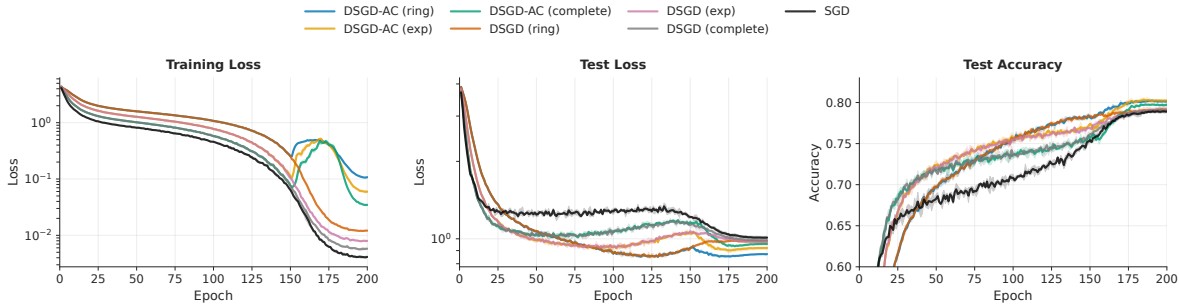

Figure 11: WRN16-8, CIFAR-100, 32 workers

### A.2.4  Sensitivity analysis

Figures 12 and 13 show the curves of the test accuracy and the test loss of the sensitivity analysis experiments.

For $p$, a greater-than-zero $p$ gives better test performance, while the benefits diminish as $p > 3$.

$E_{\text{start}}$ is the epoch from which the adaptive consensus is activated. We set $\gamma = 1$ before $E_{\text{start}}$, and make $\alpha_{\max}$ equal to the maximal learning rate among the iterations where DSGD-AC is activated.

The results show that $E_{\text{start}} = 100$ gives the best performance on test accuracy, and, for $E_{\text{start}} \in \{10, 50, 75, 100\}$, they give similar test loss performance, and a too-late activation of the adaptive consensus ($E_{\text{start}} \in \{150, 175\}$) leaves too few iterations for DSGD-AC to take effect.

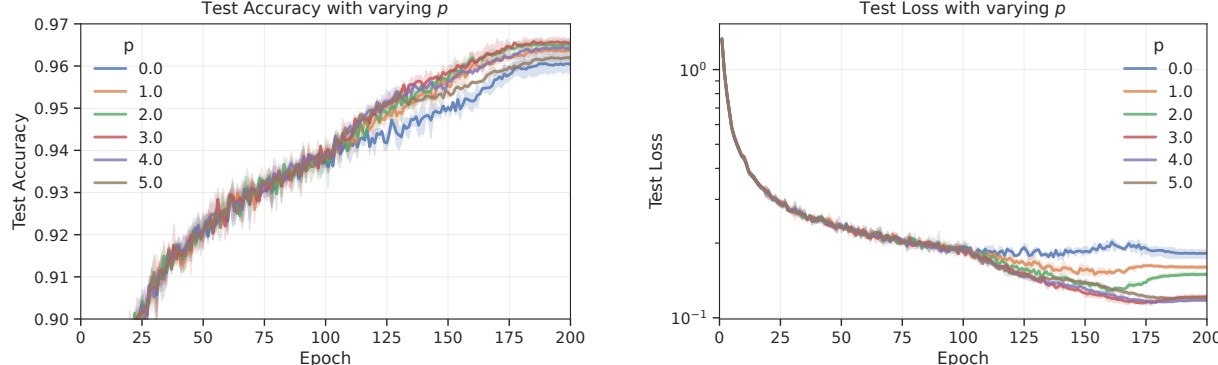

Figure 12: DSGD-AC with WRN28-10 on CIFAR-10 with varying $p$ and $E_{\text{start}} = 100$.

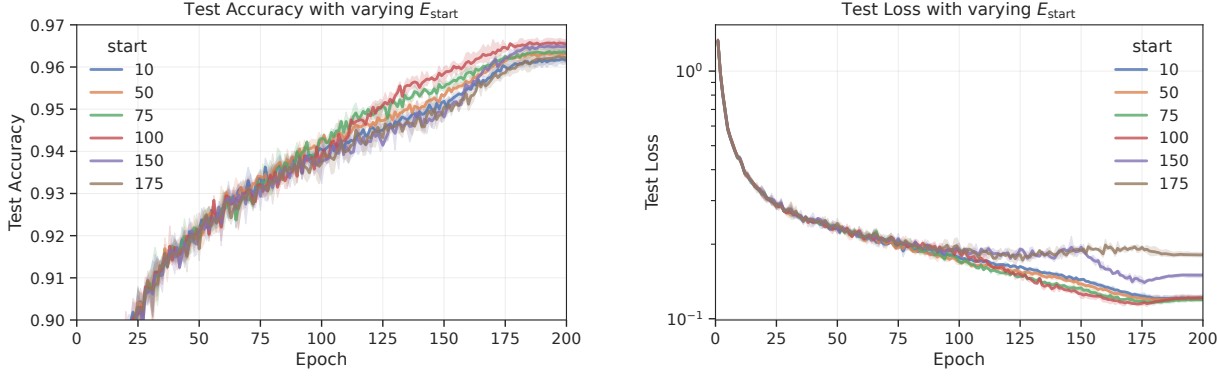

Figure 13: DSGD-AC with WRN28-10 on CIFAR-10 with varying $E_{\text{start}}$ and $p = 3$.

**Practical tuning guideline**  We use $p = 3$ as a robust initial value and first select $E_{\text{start}}$ from a coarse grid spanning the post-warm-up, middle, and late portions of training. The activation should be delayed if it causes an abrupt increase in training loss or an excessively large consensus-error norm, whereas it should be moved earlier if disagreement still collapses before adaptive consensus has sufficient time to improve validation performance. Larger worker counts and more weakly connected topologies generally warrant later candidate start epochs because they produce greater disagreement under standard DSGD. After choosing $E_{\text{start}}$, we sweep $p \in \{2, 3, 4\}$: increasing $p$ relaxes mixing more aggressively and is useful when the disagreement radius still decays, while decreasing $p$ is preferable when disagreement grows excessively or optimization deteriorates. A final local refinement of $E_{\text{start}}$ accounts for its interaction with the selected $p$. This procedure motivates

our default $p = 3$ and the setup-specific choices $E_{\text{start}} \in \{10, 100, 150\}$ for 8, 16, and 32 workers, respectively, but these epoch values should be rescaled when the training horizon or learning-rate schedule changes.

### A.2.5  Sharpness-aware minimization

We implement SAM (Foret et al., 2020) and report its results with 8 and 16 workers (1 and 2 8×T4 nodes) for reference. We follow their work to use $\rho = 0.05$ in all the experiments. The results show that SAM achieves better metric values than DSGD-AC. However, it should be noted that SAM introduces 2× computation cost as other baselines, and it is not numerically stable under mixed-precision training in our setup (the loss becomes NaN in around 30 epochs with mixed-precision training in fp16). Therefore, it takes $\sim 4.9\times$ the training time as the synchronous SGD in the 8-worker and 16-worker setups under the same number of iterations. Even under the same FLOP budget (100 epochs), SAM still takes around 2× training time, and the accuracies are comparable to DSGD-AC.

| # Nodes | Epochs | Test Acc. (%) ↑ | Test Loss ↓ | Train Loss ↓ | Training Time (min) ↓ |
|:---:|:---:|:---:|:---:|:---:|:---:|
| 1 | 200 | $83.04_{\ \pm\ 0.22}$ | $0.700_{\ \pm\ 0.007}$ | $0.0519_{\ \pm\ 0.0003}$ | $382.14_{\ \pm\ 8.32}$ |
| 2 | 200 | $83.00_{\ \pm\ 0.03}$ | $0.697_{\ \pm\ 0.004}$ | $0.0517_{\ \pm\ 0.0003}$ | $193.97_{\ \pm\ 1.18}$ |
| 1 | 100 | $82.49_{\ \pm\ 0.09}$ | $0.692_{\ \pm\ 0.002}$ | $0.0819_{\ \pm\ 0.0007}$ | $196.10_{\ \pm\ 1.33}$ |
| 2 | 100 | $82.48_{\ \pm\ 0.10}$ | $0.685_{\ \pm\ 0.005}$ | $0.0836_{\ \pm\ 0.0007}$ | $96.61_{\ \pm\ 0.78}$ |

Table 7: Performance of SAM on image classification tasks with WRN28-10 on CIFAR-100.

A mathematical comparison between SAM and DSGD/DSGD-AC is provided in Appendix A.3.4.

### A.2.6  Momentum tracking

In this section, we present the additional experiments with momentum tracking (Takezawa et al., 2022), which is a variant of gradient tracking (Nedic et al., 2017). The line of work actively promotes consensus, especially in the setup with heterogeneous data distributions across workers.

Table 8 presents the experiment results of momentum tracking with WRN16-8 and WRN28-10 on CIFAR-100 with 8 workers. In the experiments, we set $\beta = 0.9$ following the standard setup. The results show that momentum tracking can only achieve comparable performance to SGD (in Table 3) but is still outperformed by DSGD-AC by a clear margin.

| Model | Topology | Test Acc. (%) ↑ | Test Loss ↓ | Train Loss ↓ |
|:---:|:---:|:---:|:---:|:---:|
| | ring | $78.82_{\ \pm\ 0.06}$ | $1.009_{\ \pm\ 0.002}$ | $0.0043_{\ \pm\ 0.0000}$ |
| WRN16-8 | exp | $78.89_{\ \pm\ 0.26}$ | $1.008_{\ \pm\ 0.011}$ | $0.0042_{\ \pm\ 0.0002}$ |
| | complete | $78.85_{\ \pm\ 0.07}$ | $1.001_{\ \pm\ 0.005}$ | $0.0041_{\ \pm\ 0.0002}$ |
| | ring | $79.85_{\ \pm\ 0.05}$ | $1.110_{\ \pm\ 0.014}$ | $0.0023_{\ \pm\ 0.0001}$ |
| WRN28-10 | exp | $79.83_{\ \pm\ 0.16}$ | $1.093_{\ \pm\ 0.007}$ | $0.0023_{\ \pm\ 0.0000}$ |
| | complete | $80.14_{\ \pm\ 0.05}$ | $1.098_{\ \pm\ 0.024}$ | $0.0022_{\ \pm\ 0.0001}$ |

Table 8: Performance of momentum tracking with WRN16-8 and WRN28-10 on CIFAR-100 with 8 workers.

### A.2.7   Constant-learning-rate diagnostic of disagreement collapse

To isolate whether the late-stage disappearance of disagreement in standard DSGD is caused by learning-rate decay, we repeat the running WRN16-8 experiment on CIFAR-100 with 8 workers and the one-peer ring topology using a non-decaying schedule. This diagnostic, denoted by DSGD-Const, uses the same linear warm-up to $\alpha = 0.1$ over the first 10 epochs as the original experiment and then holds $\alpha = 0.1$ through epoch 200. All other optimization and communication settings are unchanged. We compare it with standard DSGD and DSGD-AC using the original cosine-decay schedule.

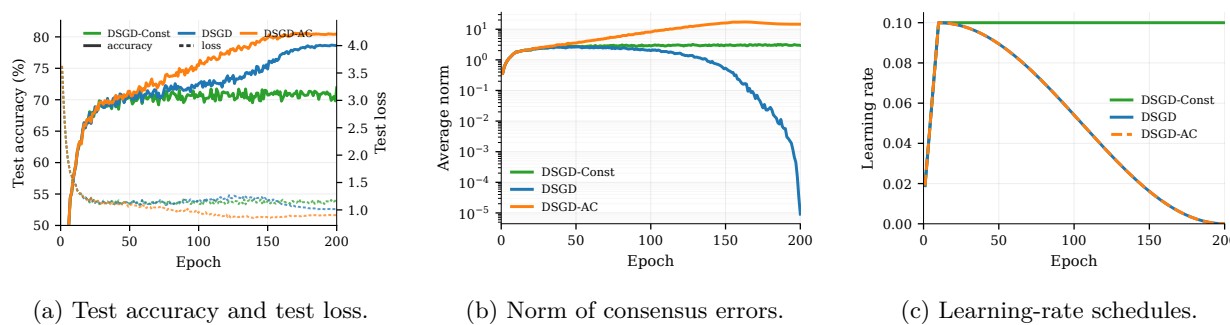

(a) Test accuracy and test loss.          (b) Norm of consensus errors.          (c) Learning-rate schedules.

Figure 14: Constant-learning-rate diagnostic for standard DSGD in the running WRN16-8 on CIFAR-100 setup. DSGD-Const keeps the learning rate at 0.1 after the 10-epoch warm-up, whereas DSGD and DSGD-AC use the same cosine-decay schedule. Panel (a) shows test accuracy on the left axis and test loss on the right axis; solid and dotted lines denote accuracy and loss, respectively. Panel (b) uses a logarithmic vertical scale.

Figure 14 shows that standard DSGD does not lose disagreement when its learning rate is held fixed. At epoch 200, the average consensus-error norm is 2.95 for DSGD-Const, whereas it falls to $9.15 \times 10^{-6}$ for DSGD with cosine decay. Thus, changing only the post-warm-up learning-rate schedule prevents the late-stage collapse by more than five orders of magnitude. Within this controlled setup, the intervention identifies learning-rate decay—which shrinks the stochastic local updates while leaving the graph-mixing contraction unchanged—as the driver of disagreement collapse, rather than collapse being an unavoidable property of standard DSGD.

As expected, removing decay degrades predictive performance: at epoch 200, DSGD-Const reaches 71.92% test accuracy and 1.1293 test loss, compared with 78.61% and 1.0143 for DSGD and 80.50% and 0.9040 for DSGD-AC[3], respectively. The constant-step control therefore verifies the causal role of learning-rate decay but is not a competitive training schedule. DSGD-AC instead retains the optimization benefit of learning-rate decay while preventing collapse by synchronizing the strength of consensus mixing with the prescribed schedule.

---

[3]These are the statistics for the running examples, which are normal to be not identical to the mean value in Table 3.

### A.3 Hyperparameter details

#### A.3.1 Hyperparameters for image classification experiments with WRN on CIFAR-10/100

The selection of hyperparameters follows the original paper (Zagoruyko & Komodakis, 2016), and our baseline implementation is consistent with its performance.

| Category | Setting |
|---|---|
| ***General*** | |
| Number of epochs | 200 |
| Global batch size | 128 for 8-worker setup and it is linearly scaled with the number of workers. |
| Learning rate scheduler | Linearly warm-up to $\alpha_{\max}$ in the first 10 epochs, followed by the cosine annealing until the end. $\alpha_{\max} = 0.1$ for 128 batch size, and it is linearly scaled with the batch size. |
| Base optimizer | SGD with momentum $\beta = 0.9$ and weight decay $5 \times 10^{-4}$. |
| Data shuffle | Randomly shuffled and split into $N$ local datasets each epoch. |
| ***Decentralized training*** | |
| Number of workers | 8, 16, and 32 |
| Communication topology | One-peer ring (alternating between neighbors $i-1$ and $i+1$ across iterations), one-peer exponential Ying et al. (2021), and complete graph (global all-reduce among all workers). |
| DSGD-AC parameters | Exponent $p = 3$; $\gamma = 1$ before $E_{\text{start}}$. |
| BatchNorm calibration | Similar to the case in Defazio et al. (2024), a calibration on the BatchNorm statistics is needed because there is a mismatch between the local models and the global average. To calibrate mismatched statistics, a full pass over the training set is conducted before validation. Only one calibration should be done if intermediate checkpoints are not evaluated. Note that we also apply the calibration to synchronous SGD for a fair comparison. |

#### A.3.2 Hyperparameters for neural machine translation experiments with Transformer on WMT14

The selection of hyperparameters follows the original paper (Vaswani et al., 2017), and our baseline implementation matches its performance. We tuned the learning rate and $(\beta_1, \beta_2)$ separately for decentralized methods.

| Category | Setting |
|---|---|
| ***General*** | |
| Number of epochs | 20 |
| Global batch size | ~50k tokens including both source and target texts |
| Learning rate scheduler | Linear warm-up to $5 \times 10^{-4}$ over the first 4000 iterations, then decay as $\alpha_0 \cdot (4000/t)^{0.5}$ ($t$ is the iteration index). $\alpha_0 = 0.0005$ for centralized Adam, and $\alpha_0 = 0.0013$ for decentralized methods. |
| Base optimizer | Adam ($\beta_1 = 0.9, \beta_2 = 0.98$) for centralized Adam, and ($\beta_1 = 0.974, \beta_2 = 0.999$) for decentralized methods. |
| Data shuffle | Randomly shuffled and split into $N$ local datasets each epoch |
| ***Decentralized training*** | |
| Number of workers | 8 |
| Communication topology | One-peer ring (alternating between neighbors $i-1$ and $i+1$ across iterations) |
| DSGD-AC parameters. | Exponent $p = 2$ (tuned based on experiments); $\gamma = 1$ during warm-up. |
| Normalization | Since only layer normalization is used, no calibration is needed. |

### A.3.3 Evaluation details

**Eigenvalue evaluation** We use the Lanczos algorithm (Lanczos, 1950) for efficient evaluation of the max and min eigenvalues of the Hessian. To be specific, the loss function is

$$F(x) = \frac{1}{B} \sum_{b=1}^{B} f(x; s_b) \tag{48}$$

where $s_b$ denotes a batch of samples from the training set, $\mathcal{S}$ denotes the total number of samples in the training set, and $\{s_1, \cdots, s_B\}$ collectively cover the entire training set. To exclude the impact of mismatched statistics in the batch normalization layers and to reflect the actual training loss, we evaluate the Hessian-vector product in training mode and control the random seed to ensure that the sample order, batching, and data augmentation are identical across all evaluations. In our evaluation, we fix the mini-batch size to 32.

We fix the number of Lanczos iterations to 30.

**Alignment evaluation** The Hessian quadratic with any vector can be computed via the dot product between the consensus error and the Hessian vector product, and the Hessian vector product (HVP) is computed using the Pearlmutter trick without materializing the Hessian, which is supported out-of-the-box by the automatic differentiation in PyTorch.

$$D := \nabla F(x)^\top v, \quad \frac{\partial D}{\partial x} = H(x)v \tag{49}$$

where $F(x)$ is defined in the same way as in Eq. (48), and, since $F(x)$ is a linear combination of loss functions on batches, the HVP can be computed in a distributed way for better efficiency.

We use 50 Hutchinson trace probes to evaluate the alignment between the Hessian and random directions in our showcase runs, and the standard deviations are on the order of $10^{-7}$.

**ISO-SGD control** ISO-SGD follows the synchronous-SGD setup of the running example and introduces Gaussian perturbations in the average-direction-SAM manner. To generate these perturbations, we record the epoch-wise average worker consensus-error norm $R_e^{\mathrm{AC}} := \frac{1}{n} \sum_{i=1}^{n} \|\delta_i^{(e)}\|$ from the DSGD-AC reference run. During epoch $e$, the sampled Gaussian perturbations are normalized to $R_e^{\mathrm{AC}}$ in the same way as Bisla et al. (2022). This gives ISO-SGD the same practical radius schedule used in Figure 1b while replacing the consensus-error directions with isotropic directions. Because the schedule is obtained from a DSGD-AC reference run, ISO-SGD is a diagnostic control rather than a deployable alternative.

**Gradient noise alignment evaluation** The gradient noises and their alignment with the Hessian are computed by

$$\xi_i := \nabla f(x; s_i) - \frac{1}{B} \sum_{b=1}^{B} \nabla f(x; s_b), \quad A_{\mathrm{grad}} := \frac{1}{B} \sum_{b=1}^{B} \xi_b^\top H \xi_b / \left( \lambda_1(H) \|\xi_b\|^2 \right) \tag{50}$$

where the first step is simply computing the difference between the mini-batch loss and the full-batch loss, and the second step is computed via HVP as in the alignment evaluation.

### A.3.4 Mathematical comparison with SAM

SAM and decentralized training can both be viewed as evaluating gradients at multiple perturbed models around a center, but they generate and control these perturbations differently. For empirical loss $\mathcal{L}_S$, the original SAM objective is (Foret et al., 2020)

$$\min_w \underbrace{\max_{\|\epsilon\|_p \leq \rho} \mathcal{L}_S(w + \epsilon)}_{\mathcal{L}_S^{\mathrm{SAM}}(w)} + \lambda_w \|w\|_2^2. \tag{51}$$

For the usual Euclidean neighborhood and mini-batch $B_{i,t}$, worker $i$ uses the first-order approximation

$$\epsilon_{i,t}^{\text{SAM}} = \rho \frac{g_{i,t}}{\|g_{i,t}\|_2}, \qquad g_{i,t} = \nabla \mathcal{L}_{B_{i,t}}(w_t), \tag{52}$$

and data-parallel SAM averages gradients evaluated at these perturbed points:

$$\widehat{g}_{\text{SAM}}(w_t) = \frac{1}{n} \sum_{i=1}^{n} \nabla \mathcal{L}_{B_{i,t}}(w_t + \epsilon_{i,t}^{\text{SAM}}), \qquad \|\epsilon_{i,t}^{\text{SAM}}\|_2 = \rho. \tag{53}$$

Thus, SAM explicitly prescribes the radius (a typical value is 0.05 (Foret et al., 2020)) and chooses each direction through an approximate inner maximization, requiring a second gradient evaluation.

For DSGD and DSGD-AC, write $x_i^{(t)} = \bar{x}^{(t)} + \delta_i^{(t)}$. Conditioned on the current worker models, the expected gradient driving the deployed average is

$$\widehat{g}_{\text{DSGD}}(\bar{x}^{(t)}) = \frac{1}{n} \sum_{i=1}^{n} \nabla f(\bar{x}^{(t)} + \delta_i^{(t)}). \tag{54}$$

Equations (53) and (54) expose the shared multi-point structure. However, the DSGD perturbations are persistent optimization states produced by stochastic-gradient differences and graph mixing; they satisfy $\sum_i \delta_i^{(t)} = 0$, need not have equal norms, and do not solve an explicit inner maximization problem.

