# OpenReview forum: "Decentralized SGD with Controlled Disagreement Finds Flatter Minima"
_TMLR — Under review for TMLR_

### Review · Reviewer_dRQV · 2026-06-30

**Summary Of Contributions:**

The paper proposes DSGD-AC, a decentralized optimization algorithm that gradually weakens consensus during training to intentionally preserve controlled disagreement among workers. The authors argue that this sustained disagreement acts as an implicit regularizer by increasing exposure to local curvature, leading to convergence toward flatter solutions and improved generalization, while incurring essentially no additional computational cost compared to standard decentralized SGD.

**Audience:**

Yes

**Audience Explanation:**

Justification-

The paper presents a simple modification to decentralized SGD that improves generalization without increasing computational complexity. So it is relevant to researchers in distributed optimization. Its central idea that controlled consensus disagreement can act as an implicit regularizer is novel and likely to stimulate further investigation.

**Broader Impact Concerns:**

The work presents a general optimization algorithm and does not raise any ethical concerns beyond those associated with machine learning in general.

**Claims And Evidence:**

No

**Claims Explanation:**

Justification-

1.⁠ ⁠The claim that $p=2$ maintains constant disagreement is stronger than the proof, which establishes constant variance only for active modes, not for total disagreement energy as the active mode set varies with $\alpha$.

2.⁠ ⁠The theoretical analysis is only for local disagreement dynamics. It does not prove $\text{larger disagreement} \;\Longrightarrow\; \text{flatter minima} \;\Longrightarrow\; \text{better generalization}$. This conclusion is rather inferred from experiments.

3.⁠ ⁠The curvature argument relies on a local quadratic approximation, neglecting $||\delta||^3$ terms. But the proposed method increases $||\delta||$ which makes the approximation less reliable.

4.⁠ ⁠Empirical evidence supports generalization by showing a correlation between preserved disagreement mechanism and improved generalization. But they do not isolate the mechanism as the reason behind it.

**Requested Changes:**

**Critical for acceptance:**

1.⁠ ⁠Clearly distinguish the results that are proved from those that are supported only by empirical evidence, particularly the connection between disagreement, flatter minima, and improved generalization.

2.⁠ ⁠Revise the statements around Corollary 2 to avoid claiming constant total disagreement, since the proof only establishes constant variance for active modes.

3.⁠ ⁠State the additional assumptions required for Corollary 1 (e.g., persistence of the active mode set) or weaken the corresponding theoretical claims.

4.⁠ ⁠Include comparisons with strong flatness-promoting baselines (e.g., SAM) to better support the claimed mechanism.

**To improve the paper:**

1.⁠ ⁠Discuss the limitations of the local quadratic/frozen-Hessian analysis and clarify the regime in which the theory is expected to hold.

2.⁠ ⁠Provide the omitted derivation showing the equivalence between the proposed surrogate objective and the DSGD update.

Overall, the work is promising. Addressing the theoretical clarifications would significantly improve the paper.

---

> ### Author Response · Authors · 2026-07-17
> **Response to Reviewer dRQV**
>
> Thank you for the precise theoretical critique and for recognizing the novelty, simplicity, and relevance of controlled disagreement as an implicit regularizer. We have uploaded a substantially revised manuscript that distinguishes proved statements from empirical findings and makes all modal claims explicitly conditional.
>
> **1. Proved versus empirical conclusions.** Section 3.4 now states that the theorem is a steady-state analysis of a frozen local stochastic window. It characterizes graph-Hessian modal variances and the balance between graph and curvature damping; it does not prove that larger disagreement globally causes flatter minima or better generalization. The abstract, Section 1, and Section 5 now describe flatter solutions and test improvements as empirical results in the evaluated settings. Section 5 and Appendix A.1.7 explicitly state that the analysis provides no global convergence or generalization guarantee and does not model changing Hessians, learning rates, or noise covariances. We listed them as explicit limitations.
>
> **2. Constant variance under $p=2$.** Corollary 2 in Section 3.4, with proof in Appendix A.1.6, now states that $p=2$ gives constant-order variance only for each mode that remains active and mixing-dominated. Because the active condition tightens as $\alpha$ decreases, it no longer claims constant total disagreement; the possible growth and instability for $p>2$ are also stated.
>
> **3. Additional conditions for the radius claim.** Corollary 1 in Section 3.4, with proof in Appendix A.1.5, introduces the active low-curvature set and its noise-weighted mass $M_{\mathrm{low}}(\alpha,\gamma)$. Its aggregate contribution is $\Theta((\alpha^2/\gamma)M_{\mathrm{low}})$ and reduces to $\Theta(\alpha^2/\gamma)$ only when this mass remains bounded above and below by positive constants. The stability and persistent-noise requirements are now explicit.
>
> **4. Strong flatness-promoting baseline.** Section 4.3 surfaces the SAM results from Appendix A.2.5: full-budget SAM attains better metrics at about 4.9x the wall-clock time, while matched-FLOP SAM has comparable accuracy but remains about 2.4x slower. Appendix A.3.4 mathematically contrasts SAM's fixed-radius inner maximization with DSGD/DSGD-AC's endogenous zero-sum perturbations. Section 3.3 (Figure 3) additionally provides the radius-matched ISO-SGD control. Taken together, radius preservation is the primary driver of maintained curvature exposure, and the ISO-SGD experiments show that structured directions contribute an additional benefit beyond radius alone.
>
> **5. Limitations of the local quadratic analysis.** Section 3.3 acknowledges that higher-order terms may not be negligible at the DSGD-AC radius. Appendix A.2.1 evaluates the exact aggregate loss along observed deviations. The quadratic residual is non-monotone (13.4% at epoch 100, 46.2% at epoch 160, and 18.3% at epoch 200), and the odd-order contribution is only 1.6% at the final checkpoint. We therefore use the Hessian quadratic only as a local curvature-exposure diagnostic, not as an exact finite-radius loss envelope.
>
> **6. Omitted surrogate derivation.** We add Appendix A.1.1 that derives the surrogate explicitly.

---

### Review · Reviewer_imZ1 · 2026-07-05

**Summary Of Contributions:**

The paper tackles the issue of disagreement, or consensus error, in decentralized SGD (DSGD). Namely, this paper aims to disprove the notion that disagreement results in underperformance. They do this in two main strands.

First, theoretically, the paper shows that disagreement acts as an implicit sharpness-aware regularizer that is able to find flatter minima whilst still having a directionality component, which means that DSGD achieves curvature-dependent damping that suppresses sharp directions while maintaining error magnitude along flatter ones. Empirically, they show this effect on image classification benchmarks where decentralized SGD, with a certain amount of disagreement maintained, is able to find flatter solutions with lower top Hessian eigenvalues and outperform normal DSGD and centralized SGD baselines.

The caveat in their paper, however, is that this disagreement interacts adversely with the learning rate schedule. Specifically, as the learning rate $\alpha^{(t)}$ decays toward zero late in training, the effective consensus regularizer weight diverges ($1/(4\alpha^{(t)})$), which diminishes the disagreement. As such, they introduce a novel scheduling mechanism called Adaptive Consensus (DSGD-AC). This applies a scaling factor ($\gamma^{(t)} = [\alpha^{(t)}/\alpha_{\max}]^p$) to the consensus portion of the model update step per-worker to prevent the learning rate schedule from diminishing the disagreement, despite its proved importance, as training progresses.

**Strengths:**

The paper provides solid theoretical backing for why consensus errors act as a beneficial sharpness regularizer, and offers a principled mitigation for how standard learning rate schedules diminish this effect.

**Weaknesses:**

The paper relies too heavily on standard SGD and decaying learning rate schedules for its narrative framing. The learning rate schedule interaction is merely a caveat that would not appear should other schedules be used. Furthermore, the core phenomenon applies broadly across other optimizers, and it should be reflected more strongly throughout the paper.

**Additional Comments:**

n/a

**Audience:**

Yes

**Audience Explanation:**

Briefly, the core theoretical contribution in Section 3.4 is convincing in showing the benefits of disagreement with respect to optimization performance and generalization. This, coupled with the empirical evidence, provides an informative resource for TMLR's audience.

**Claims And Evidence:**

No

**Claims Explanation:**

While my overall assessment leans negative regarding the completeness of the empirical evidence, the paper has partially fulfilled what is needed to support its central claims. First, it is worth noting that the theoretical component is strong. It clearly elucidates why controlled disagreement acts as an implicit sharpness regularizer in DSGD and why preserving it can lead to flatter minima and better generalization. However, the empirical evaluation has two gaps that prevent me from fully validating their overarching claims:

1. While the authors compare against standard DSGD and centralized SGD, the paper does not compare against decentralized baselines that explicitly attempt to eliminate or correct disagreement. Although the authors reference this body of literature, omitting empirical comparisons against these methods leaves doubt as to whether intentionally maintaining disagreement (via DSGD-AC) is genuinely superior to the alternatives that they have mentioned in the paper.
2. The empirical and theoretical motivation for Adaptive Consensus (AC) is tightly coupled to standard asymptotic learning rate decay. If the paper had evaluated alternative modern schedules (e.g. WSO) or constant-learning-rate regimes, the consensus radius might not collapse prematurely, potentially rendering the proposed AC mechanism unnecessary. Because the authors do not evaluate or discuss how their claims hold under alternative scheduling paradigms, framing DSGD-AC as a universal mechanism rather than a scheduler-specific compensation technique makes the paper's core claims overly strong.

**Requested Changes:**

Below, I suggest adjustments to the work, where I indicate its severity level accordingly:

1. Provide experimental comparisons or a rigorous theoretical analysis contrasting DSGD-AC against exact or consensus-corrected decentralized baselines. Demonstrating that intentionally maintaining controlled disagreement outperforms state-of-the-art decentralized algorithms designed explicitly to eliminate disagreement is essential to substantiate the overarching claim that disagreement inherently benefits optimization and generalization. [Critical]

2. Reframe the ordering of Section 3 by placing Section 3.4 earlier (ideally right after Section 3.1). Presenting the theoretical modal explanation first provides a much stronger conceptual foundation for why controlled disagreement is desirable. Furthermore, this structural adjustment reduces narrative reliance on the specific cosine learning rate schedule, highlighting that the underlying principles apply broadly across alternative scheduling regimes. [Critical]

3. Reorder the contributions throughout the overarching narrative flow of the paper. The theoretical framing should be presented first to establish the intrinsic benefits of disagreement, followed by the adaptive consensus mechanism as a practical intervention to mitigate learning rate decay, and concluding with the empirical evaluations. [Critical]

4. Move the experimental results involving alternative optimizers (such as Adam/AdamW from the appendix) into the main text. Demonstrating that consensus-error regularization holds across different optimizer mechanics significantly strengthens the claimed generality of the proposed method. [Critical]

5. Moderate the framing surrounding the "Adaptive Consensus" mechanism. Because the scaling factor is strictly deterministic and tied directly to a predefined learning rate schedule, describing the mechanism as "adaptive" is somewhat misleading. Re-labeling or reframing it as a schedule-synchronized consensus scaling mechanism will better reflect its actual implementation. [Critical]

6. To further isolate the causal mechanism behind disagreement collapse, please provide diagnostic experiments with non-decaying learning rate schedules. While I acknowledge that eliminating learning rate decay will degrade performance, this would verify whether the late-stage disappearance of disagreement is strictly an artifact of a decaying learning rate. [Critical]

7. Revise the abstract to provide clearer narrative progression and better contextualize the reader. Currently, the transition from the problem setting to the proposed contribution is abrupt; the abstract should explicitly state why and how consensus error acts as a beneficial sharpness regularizer before introducing the proposed mechanism. [Critical]

8. Provide experimental ablations that systematically vary the baseline level of inherent disagreement (e.g., by altering communication topology, $W$). While the paper focuses on preventing late-stage disagreement collapse, it does not control the amount of disagreement occurring in orthogonal aspects of the work. Investigating varying baseline levels of disagreement is necessary to confirm whether maintaining it is strictly advantageous across diverse training regimes. [Critical]

9. Clarify the necessity and role of $\alpha_{\max}$ in Algorithm 1. While $\alpha_{\max}$ is introduced to ensure the consensus scaling factor remains bounded within $[0, 1]$, it is omitted in the theoretical analysis of Section 3.4. Explicitly distinguishing its role as a practical normalization factor to prevent over-mixing from the theoretical stability conditions would prevent reader confusion. [Beneficial for clarity]

10. Include an explicit mathematical comparison with Sharpness-Aware Minimization (SAM), potentially in the appendix. Contrasting SAM's fixed perturbation radius directly against the modal consensus-error radius of the proposed approach would provide clearer theoretical intuition for readers. [Beneficial for clarity]

11. Could the authors provide a concrete heuristic or recommended search guideline for $p$ (for instance, a simple grid search starting from $p=2$ upwards in integer increments)? Furthermore, briefly discussing whether the optimal choice of $p$ is expected to transfer reliably across different architectures, datasets, and network topologies would significantly aid practitioners seeking to adopt or reproduce this method. [Beneficial for clarity]

12. Please can the authors provide citations to back the claim: “widely regarded as inferior to centralized training….” [Critical for context]

---

> ### Author Response · Authors · 2026-07-17
> **Response to Reviewer imZ1**
>
> Thank you for the detailed and constructive review. We have uploaded a revised manuscript addressing the concerns.
>
> **1. Consensus-promoting baseline.** Section 4.3 and Appendix A.2.6 add momentum tracking [1], an variant of gradient-tracking method [2] that actively promotes consensus, on WRN16-8 and WRN28-10 across topologies. It remains near SGD/DSGD and is consistently outperformed by DSGD-AC in the evaluated i.i.d. setting. Because gradient tracking primarily targets heterogeneous data, behavior close to standard DSGD is expected here. The comparison directly tests suppressing versus maintaining disagreement rather than the strength of a heterogeneity correction.
>
> **2. Ordering of Section 3.** We respectfully retain the current order: Section 3.1 establishes decay-induced disagreement collapse; Section 3.2 defines DSGD-AC; Section 3.3 separates radius, direction, and curvature exposure; and Section 3.4 explains those quantities through local modal theory. Moving Section 3.4 earlier would introduce graph-Hessian modes before the algorithm and empirical quantities that motivate them, making the presentation less accessible.
>
> **3. Overall narrative order.** We agree that the reader should understand why disagreement can be beneficial before the algorithm. The new opening of Section 3 explains qualitatively that stochastic updates create disagreement, mixing removes it, and learning-rate decay tilts this balance toward collapse. The rewritten abstract and introduction also present the graph-curvature damping mechanism early. We retain only the formal modal machinery after Section 3.3 because Lemma 1 and Theorem 1 use the empirical quantities $Q_t$, $A_t$, and $B_t$ defined there.
>
> **4. Alternative optimizers.** Section 4.3 now surfaces the DAdam-AC result from Appendix A.2.2. Since adaptive preconditioning changes the noise geometry used by our analysis and the empirical results show relatively smaller improvement than in image classification, we present this as preliminary compatibility evidence rather than proof of optimizer-independent generality.
>
> **5. Meaning of "adaptive."** Section 3.2 consistently describes $\gamma^{(t)}$ as **learning-rate-synchronized consensus scaling** and clarifies that "adaptive" means deterministic adaptation to a predetermined schedule, not feedback from an observed training statistic. The method is explicitly scoped to decaying-learning-rate regimes.
>
> **6. Non-decaying schedule diagnostic.** Section 3.1 and Appendix A.2.7 (Figure 14) report DSGD-Const with all other settings unchanged. Its final disagreement norm is 2.95, compared with $9.15\times10^{-6}$ for cosine-decay DSGD, confirming that decay drives the collapse in this setup. However, its accuracy is only 71.92%. DSGD-AC therefore retains the optimization benefit of decay while compensating for its disagreement-collapse side effect.
>
> **7. Abstract.** The abstract now states the local frozen-window scope, explains the magnitude/direction mechanism before the empirical conclusion, and separates the theoretical characterization from the observed performance improvements.
>
> **8. Baseline disagreement across topologies.** Section 4.2 evaluates 8, 16, and 32 workers with ring, exponential, and complete topologies. Appendix A.2.3 explains DSGD-AC's greater topology sensitivity through the graph spectrum in the modal variance. DSGD-AC nevertheless outperforms DSGD across every evaluated topology.
>
> **9. Role of $\alpha_{\max}$.** Section 3.2 defines $\alpha_{\max}$ as a practical normalization that keeps $\gamma^{(t)}\in[0,1]$ and prevents mixing stronger than base DSGD; it is not an additional theoretical stability assumption.
>
> **10. Mathematical and empirical comparison with SAM.** Section 4.3 surfaces the SAM results from Appendix A.2.5. Full-budget SAM attains better metrics but takes about 4.9x the wall-clock time; matched-FLOP SAM has comparable accuracy but remains about 2.4x slower. Appendix A.3.4 contrasts SAM's prescribed-radius inner maximization with DSGD/DSGD-AC's endogenous zero-sum deviations.
>
> **11. Tuning guideline for $p$.** Section 4.1 and Appendix A.2.4 recommend starting with $p=3$, selecting $E_{\mathrm{start}}$ on a coarse temporal grid, sweeping $p\in\{2,3,4\}$, and refining the start epoch while monitoring disagreement and optimization. The guideline discusses worker count, topology, training horizon, and schedule.
>
> **12. Supporting citations.** Section 1 now cites prior work supporting the statement.
>
> [1] Takezawa, Yuki, et al. "Momentum tracking: Momentum acceleration for decentralized deep learning on heterogeneous data." arXiv preprint arXiv:2209.15505 (2022).
>
> [2] Nedic, Angelia, Alex Olshevsky, and Wei Shi. "Achieving geometric convergence for distributed optimization over time-varying graphs." SIAM Journal on Optimization 27.4 (2017): 2597-2633.

---

### Review · Reviewer_xJU6 · 2026-07-07

**Summary Of Contributions:**

**Summary**:
This paper proposes a novel algorithm, named DSGD-AC, for decentralized training. The algorithm uses a time-dependent scaling mechanism to maintain consensus errors throughout training. The authors claim that this adaptive consensus algorithm can find flatter minima and improve generalization, while incurring minimal computational cost compared to traditional sharpness-aware optimizers such as SAM. In addition, they provide theoretical analysis on modal variance and consensus-error radius, together with empirical diagnostics. Experiments on image classification support the effectiveness of the proposed algorithm.

**Strengths:**
1. The motivation is clear. The authors build on the work of Zhu et al. (2023) and observe that the consensus error, viewed as an implicit sharpness-aware regularizer, vanishes during the optimization process. They further argue that this consensus error should not vanish, which serves as the main motivation of the paper.
2. The algorithm is very simple and clean compared to the DSGD baseline. It only introduces one additional parameter $\gamma_t$, to control the consensus regularizer.
3. The theoretical analysis, together with the empirical diagnostics, is reasonably thorough. It contains a variety of analyses supporting the proposed mechanism.
4. The experimental section includes extensive studies, including different hyperparameter settings, ablation studies, sensitivity analysis, and training efficiency.

**Weaknesses:**
1. The main focus of this paper is the role of consensus error in the decentralized setting. The authors argue that, instead of reducing these errors as in previous decentralized optimization methods, they intentionally maintain the disagreement. However, this argument is not yet fully convincing, and several questions remain (see Requested Changes for more details). In addition, if the consensus error is intentionally maintained, how would the convergence behavior of DSGD-AC differ from that of DSGD? Would the algorithm still converge theoretically?
2. The theoretical analysis relies on relatively strong assumptions, and there is an explicit gap between the theory and the experiments. In particular, the frozen Hessian and constant modal noise variance assumptions are strong, and it remains unclear whether they hold in the deep learning experiments. Moreover, Theorem 1 assumes fixed $\alpha$ and $\gamma$, whereas both parameters are time-dependent in the algorithm and experiments. Although the authors mention "local windows," the connection between the stationary analysis and the actual training trajectory remains loose.
3. Some experimental results are not sufficiently discussed. For example, in Table 1, increasing parameter $p$ leads to higher training loss (both mean and standard deviation) but lower test loss, whereas in Table 2, increasing $E\_{start}$ leads to lower training loss but higher test loss. How can these observations be explained? Moreover, in Figures 6-11, especially in the 8-worker setting, DSGD appears to be more robust across different communication topologies than DSGD-AC. How do the authors explain this phenomenon?

**Audience:**

Yes

**Audience Explanation:**

Decentralized optimization is an active research area, and this paper challenges the traditional view of consensus error. The actual role of consensus error remains unclear in the existing literature, and this paper provides an interesting perspective by arguing that maintaining consensus errors, rather than reducing them as in previous work, may be beneficial for decentralized training. In addition, the paper is also related to sharpness-aware optimization (e.g., SAM) and generalization analysis, although the discussion of SAM is more of an extension of the work by Zhu et al. (2023).

**Broader Impact Concerns:**

The paper already includes a Broader Impact Statement.

**Claims And Evidence:**

No

**Claims Explanation:**

If one considers improved test accuracy, lower test loss, and smaller dominant Hessian eigenvalues, the proposed DSGD-AC does show some benefits. However, the paper's main claim is still not fully supported. The role of consensus error with respect to convergence, loss landscape, and disagreement remains unclear. In addition, there is a noticeable gap between the theory and practice. The theoretical analysis assumes a frozen Hessian and constant learning rate, whereas the experiments employ cosine learning-rate decay and warm-up schedules.

**Requested Changes:**

**Critical**
1. The explanation around Figure 3 could be strengthened. In Figure 3(b), the authors argue that the consensus errors of both DSGD and DSGD-AC are not random but instead exhibit meaningful alignment with the Hessian. However, the figure also shows that DSGD consistently achieves higher normalized alignment than DSGD-AC during much of training. Moreover, Figure 1(b) suggests that DSGD also maintains relatively large consensus errors during the early stages of training. This raises an important question regarding the proposed mechanism. If DSGD exhibits both stronger alignment and substantial disagreement early in training, why does DSGD-AC ultimately converge to flatter minima and achieve better generalization? The manuscript argues that the larger disagreement radius of DSGD-AC compensates for its weaker alignment through increased curvature exposure, but this explanation does not fully disentangle the respective roles of disagreement magnitude and disagreement direction. Could the authors provide additional discussion or analysis to clarify whether the observed improvements arise primarily from where the disagreement points (i.e., its interaction with the Hessian), or simply from maintaining a larger disagreement radius?
2. It would be helpful to include additional discussion, preferably with theoretical support, on the relationship between consensus error and convergence. For example, could maintaining non-vanishing consensus errors lead to slower convergence or even divergence?
3. The theoretical analysis should be strengthened. In particular, the authors should discuss whether the assumptions can be relaxed. If not, they should explain whether these assumptions are reasonable in practical deep learning settings. In addition, the gap between Theorem 1 and the practical algorithm should be better discussed.

---

> ### Author Response · Authors · 2026-07-17
> **Response to Reviewer xJU6**
>
> Thank you for the careful review and for highlighting the need to separate disagreement magnitude, direction, and theoretical scope. We have uploaded a revised manuscript that addresses these points.
>
> **1. Disagreement radius versus direction.** Section 3.3 now decomposes curvature exposure exactly as
> $Q_t=\|\Delta^{(t)}\|_F^2 A_t$, where $\|\Delta^{(t)}\|_F$ is the disagreement radius and $A_t$ is normalized Hessian alignment. We explicitly acknowledge that standard DSGD often has stronger normalized alignment and substantial early disagreement. Its radius nevertheless collapses under learning-rate decay, causing late-stage curvature exposure to collapse. DSGD-AC preserves a much larger radius while retaining above-isotropic directional structure.
>
> In Section 3.3, We also added ISO-SGD, a radius-matched isotropic control, in Figure 3. It recovers much of the accuracy gain, confirming that magnitude matters, but reaches a substantially sharper and slightly less accurate solution than DSGD-AC. Thus, radius preservation is the primary driver of maintained curvature exposure, and ISO-SGD shows that structured directions contribute an additional benefit beyond radius alone.
>
> **2. Convergence and theoretical scope.** We added Appendix A.1.7 that relates maintained disagreement to standard DSGD stationarity analyses. These bounds contain the term $\frac{C_{\Delta}}{T}\sum_{t=1}^T\mathbb{E}[\|\Delta^{(t)}\|_F^2]$. If $ |\Delta^{(t)}|_F\leq R $, the average model approaches a wider stationarity neighborhood rather than exact consensus. If the radius is unbounded, the bound becomes vacuous and divergence is possible. Our finite-horizon runs maintain bounded disagreement, while overly aggressive preservation increases training loss. This is a standard-bound implication plus finite-horizon evidence, not a new global convergence theorem.
>
> Section 3.4 further limits the local theory's scope. Theorem 1 establishes each graph-Hessian mode's unique steady-state second moment only when $0<\gamma\mu_j+\alpha\lambda_k<2$. A nonstationary trajectory need not track this state, and the analysis provides neither a global convergence nor a generalization guarantee. Outside this interval, a mode is not mean-square contractive.
>
> The constant-learning-rate diagnostic in Appendix A.2.7 further clarifies the trade-off. With all other settings unchanged, a non-decaying step retains a final disagreement norm of 2.95 instead of $9.15\times10^{-6}$ under cosine decay, but accuracy falls to 71.92%, versus 78.61% for DSGD and 80.50% for DSGD-AC. Persistent disagreement alone is therefore insufficient; DSGD-AC preserves controlled late-stage disagreement while retaining the optimization benefit of decay.
>
> **3. Assumptions and relaxation.** The revised corollaries in Section 3.4 are explicitly conditional on stable active modes and persistent noise-weighted modal mass. For $p=2$, constant-order variance is claimed only for modes that remain mixing-dominated, not for total disagreement. The tightening active-mode condition and the risk of growth for $p>2$ are now stated.
>
> The frozen-window assumptions can be partially relaxed by replacing constant $\alpha$ and $\gamma$ with slowly varying schedules, but this yields less transparent tracking statements rather than the closed-form steady state. We therefore state Theorem 1 as a frozen-model damping comparison, not a trajectory guarantee. Allowing the Hessian and noise covariance to vary freely would additionally couple modes and require a genuinely time-varying analysis, which we leave for future work. Empirically, Appendix A.2.1 tests the local quadratic model along observed deviations. Its residual is non-monotone (13.4% at epoch 100, 46.2% at epoch 160, and 18.3% at epoch 200), so we claim local adequacy near the final solution rather than throughout training; the final odd-order contribution is only 1.6%. Figure 2c also shows stable late-training gradient-noise structure over the evaluation window, supporting the local constant-noise idealization. These measurements support the intended local late-training use without claiming that the frozen approximation holds globally.
>
> **4. Practical behavior.** Section 4.1 and Appendix A.2.4 expand the sensitivity discussion. Increasing $p$ weakens mixing more aggressively, raising disagreement and training loss while improving test performance until excessive disagreement or alignment degradation reverses the benefit. Delaying $E_{\mathrm{start}}$ lowers training loss but leaves less time for regularization, increasing test loss when activation is too late. More workers and weaker connectivity generally require later activation. Section 4.2 and Appendix A.2.3 explain the larger topology-dependent spread through the graph spectrum; DSGD-AC still outperforms DSGD across every evaluated topology. Appendix A.2.4 also gives practical tuning guidance.